# Risk Analysis on PMMA Recycling Economics

**DOI:** 10.3390/polym13162724

**Published:** 2021-08-15

**Authors:** Jacopo De Tommaso, Jean-Luc Dubois

**Affiliations:** ARKEMA France, 420 Rue d’Estienne d’Orves, 92705 Colombes, France; jacopo.de-tommaso@polymtl.ca

**Keywords:** methyl methacrylate, PMMA recycling, Monte Carlo, economic analysis, regenerated MMA, depolymerization, scenario, net present value (NPV), payback period, risk analysis

## Abstract

Poly(methyl methacrylate) (PMMA) is a versatile polymer with a forecast market of 4 Mtons/y by 2025, and 6 USD billion by 2027. Each year, 10% of the produced cast sheets, extrusion sheets, or granules PMMA end up as post-production waste, accounting for approximately 30 000 tons/y in Europe only. To guide the future recycling efforts, we investigated the risks of depolymerization process economics for different PMMA scraps feedstock, capital expenditure (CAPEX), and regenerated MMA (r-MMA) prices via a Monte-Carlo simulation. An analysis of plastic recycling plants operating with similar technologies confirmed how a maximum 10 M USD plant (median cost) is what a company should aim for, based on our hypothesis. The capital investment and the r-MMA quality have the main impacts on the profitability. Depending on the pursued outcome, we identified three most suitable scenarios. Lower capital-intensive plants (Scenarios 4 and 8) provide the fastest payback time, but this generates a lower quality monomer, and therefore lower appeal on the long term. On 10 or 20 years of operation, companies should target the very best r-MMA quality, to achieve the highest net present value (Scenario 6). Product quality comes from the feedstock choice, depolymerization, and purification technologies. Counterintuitively, a plant processing low quality scraps available for free (Scenario 7), and therefore producing low purity r-MMA, has the highest probability of negative net present value after 10 years of operation, making it a high-risk scenario. Western countries (especially Europe), call for more and more pure r-MMA, hopefully comparable to the virgin material. With legislations on recycled products becoming more stringent, low quality product might not find a market in the future. To convince shareholders and government bodies, companies should demonstrate how funds and subsidies directly translate into higher quality products (more attractive to costumers), more economically viable, and with a wider market.

## 1. Introduction

Poly(methyl methacrylate) (PMMA), known as acrylic or acrylic glass, as well as Perspex, Plexiglas, and Altuglas, is a transparent thermoplastic and an alternative to glass due to its UV resistance and transparency.

The methyl methacrylate (MMA) global market was valued 7.8 billion US dollars in 2018 [1], and the production volume assessed around 3.9 Million tons in 2019/2020 [2,3]. These figures are foreseen to increase steadily in the next years, to reach 5.7 Million tons of MMA worldwide produced by 2028 [2], and an estimated worth of 4.8 USD billion for MMA by 2028 in Europe only. Similarly, the PMMA market volume passed from 2 Million tons/y in 2017 [4], to 2.6 Million tons in 2019 [5], and it is estimated to reach a market value of 6 USD billions by 2027 [6].

The Covid 2019 pandemic had a significant effect on the PMMA market. On one side, the global crisis put a strain on the automotive and construction industry, that accounts for more than 60% of the PMMA application [3,7]. At the same time, sheets production boomed, along with demand for medical diagnostic equipment such as incubators, or medical cabinets, among others. For this reason, although the production of PMMA incidentally decreased in the first two quarters of 2020, spot prices and requests for PMMA increased by 25% in 2020 [7], making it the most sought-after polymer in the world. In particular, the protective sheet market made the feedstock price (MMA) bouncing back after 27 months of pricing decline [8], and therefore ICIS foresees a “strong (+10%) rebound in 2021” (of PMMA market), “to be completed in 2022 returning to pre-crisis demand levels” [3].

Concurrent with the increasing market and production volume trend of the last years, the End-of-Life and post-production waste ramped up as well. The MMAtwo consortium estimated that, in Europe only, out of the 300,000 tons/y produced, only 8000 are recycled in Europe, most of which (70%) coming from post-production streams such as PMMA offcuts, saw dust, production scraps, or off-grades [9].

The market demands of PMMA and MMA represent a relatively small volume compared to other plastics (10–100 times less [10,11]). Therefore, the general public often oversees strength and opportunities of PMMA recycling. Nevertheless, already back in 1945, only 17 years after Otto Röhm first synthesized PMMA, Gem Participations patented a “heat transfer bath process” to depolymerize “Lucite” scrap back to the monomer [12]. Due to its higher price (compared to large volume plastics like polyolefins for instance), several companies and academic groups investigated PMMA chemical recycling to MMA. PMMA thermally depolymerizes to MMA by radical unzipping starting at 350 °C, and pure PMMA fully depolymerizes to MMA at 450 °C [13]. Thermochemical technologies emerged in time, with the principal being dry distillation, molten metal bath, fluidized beds, and extruders [10,14]. Regardless the technology, the economic viability has always hurdled the large-scale industrialization of any recycling process [15]. To guide the future recycling efforts, we investigate the economics of PMMA recycling, detailing the effect of feedstock type, product quality, and capital investment.

The “Dry-distillation” is commonly practiced in several countries, including India, China, Brazil… In this process, PMMA scraps are loaded in a tank, which is heated with a direct burner most of the time, and when no more cracked products are emitted, the unit is cooled to be reloaded. Crude MMA is condensed, and purified, most often with a rough distillation. The “Rotating-drum” process is a variation of the dry-distillation, which is also popular in Asia. It can be operated continuously, but most of the time it is a batch process. The PMMA scraps are loaded in the rotating cylinder, which provides a kind of mixing of the material, and avoids one of the drawbacks of the dry distillation (the heat transfer limitation through the char layer which forms in the unit). The “Molten Metal” process, uses most often molten lead, but can also use tin or zinc. The molten metal has a high specific heat which is efficiently transferred to the PMMA. When these units are operated properly there is no contamination of the product nor the environment. However, when low quality scraps are processed, a high amount of solid residues are produced, contaminated with the metal used. This generates additional cost to dispose them of properly. This technology is still in operation in several countries, including Italy, Spain, Egypt, and in South East Asia for example. The “Fluid-Bed” process involves a twin fluid bed circulating reactor system. Hot sand is circulated between a depolymerization reactor and a regenerator where carbon deposits are burned and sand is reheated. This process has been piloted at rather large scale, but was not industrialized yet, although several variations have been investigated. Other technologies are under development like the use of microwaves, Joule effect (ohmic heating), and inductive heating. “Stirred-Tank” process is also a variation of the dry distillation, in which a stirrer is used in the reactor for better heat and mass transfer. That requires a molten polymer base to be able to stir the scraps. To the best of our knowledge, a few units are using this type of technology. The technology investigated in the MMAtwo research project is based on a “Twin-screw extruder”. This process was used at industrial scale on high-quality cast PMMA scraps, and is investigated in the project for all kinds of PMMA wastes, including post-industrial and post-consumer wastes. In this process, the scraps are continuously fed in the extruder where the residence time is very short, and the crude MMA vapors are condensed, and further purified.

All the processes, in use today, include a scrap pretreatment (crushing, separation of the polyethylene films and PVC parts), a condensation, and a purification section. The level of purification varies greatly between the sites, with product as low as 91 wt % MMA content seen and up to 98.5 wt % seen on the market, or 99.8 wt % achieved in the MMAtwo project. The sequence of purification steps depends on the usual impurities and the target for the regenerated MMA producer. The distillation can be operated in a batch mode with a single column or in a continuous mode with two distillation units, and in some cases with extra purification steps such as washing steps and dehydration steps. So, the capital cost and operating cost are strongly affected by the level of purity expected for the r-MMA. At the same time, the quality of the scraps has also a strong impact on the final r-MMA purity: a clean cast PMMA will give better r-MMA quality as it does not contain co-monomers unlike the injection and extrusion grades PMMA. Therefore, not all the scraps have the same market value.

## 2. Materials and Methods

### 2.1. Depolymerization Process

At a first glance, typical PMMA recycling processes have a common backbone (Figure 1):Pre-treatment in or off-site (1000);Depolymerization section (2000);Purification section (3000).

The nature of each process step varies based on the interdependence of feedstock source, the target product quality, and the available technological know-how. Combination of different steps gives the full spectrum of process technologies (dry distillation, molten metal bath, fluidized bed, extruder, etc.,).

Although no process route is dominating worldwide [10], some are preferred when lower quality r-MMA is acceptable, or less capital-intensive plants are required. For instance, dry distillation plants have a low capex, but produce a lower quality monomer (<98% wt.), nevertheless accepted in the local markets (e.g., India, China, Brazil) [10,16,17]. Molten lead baths, instead, are very sensitive to PMMA scraps quality, and have a proven track record with high-quality waste [10,18]. The MMAtwo twin-screw extruder technology depolymerized several post-industrial, and end-of-Life PMMA grades, into a r-MMA as pure as 99.8 wt. % (after distillation) [19]. However, this comes with a higher capital investment, compared to dry distillation, for instance. Theoretically, the ideal technology not only produces the best product from the worst waste, at the lowest capital investment, but it also eases as much as possible its environmental footprint. For this reason, we report the flowsheet of our own PMMA design of recycling process, organized here to be a compromise or average representative of the existing technologies and heavily energy integrated (Figure 1 and Figure 2).

The PMMA scraps are pre-treated in or off site (section 1000), crushed, granulated, and then conveyed to the reaction section (2000). Because it is nowadays seen more and more often in mixed waste plastic recycling technologies, and because it was explored in the past by Mitsubishi Rayon, [20] we considered a stirred tank reactor for the depolymerization. The PMMA granules are pre-melt (M-2001), for instance in a single screw extruder, and then depolymerized in a stirred tank reactor R-2001 (Figure 1). This combination of melter/reactor is similar to what we see in Agilyx, or plastic energy [21,22] recycling technologies. Compared to a “typical” PMMA process, we minimized the capital and operating expenses (Capex/Opex) at the pre-treatment by not including a washing step.

The depolymerization vapors condense and are then purified to r-MMA in a series of columns. First, the crude is pre-purified from the very heavy ends in the evaporator F-3001, and then two distillation columns fix the purity of the monomer. The plant produces by-products, burnt on site for energy recovery (section 4000—not shown), r-MMA, and solid and liquid waste (heavy ends), which are disposed of outside the battery limits. The pigments, fillers, charred polymer, and other polymer residues tend to accumulate in the solid residue, which might also contain some substances of very high concern, therefore requiring proper disposal protocols. The heavy ends contain dimers and oligomers, but also some other cracked polymers that contaminated the PMMA.

Reaction and purification sections are technology-neutral, meaning that any other reactor or purification choice are valid in lieu of our design, since our goal is to explore the economics of PMMA recycling at large. Same applies for the heating medium choice (molten salt or hot oil), that are interchangeable with any other heating carrier (e.g., steam, flue gases, or direct heating) as long as we assume the same heating energy demand. Other purification technologies are sometimes used by operators such as a washing step with sodium hydroxide solution, but that introduces water in the MMA, and increases the complexity of the distillation due to an azeotrope between water and acrylates and methacrylates. In absence of added water washing step, the losses due to the azeotropes can be minimized.

The by-products combustion provides heat energy and steam for the rotative equipment. Natural gas complements the heating duty required. Natural gas and by-products fuel a molten salt, and a hot oil circuit. The molten salt loop supplies the heat to the reaction section, while a hot oil circuit supplies the purification section. Because of the high energy content of the by-products (comparable to that of MMA—26.2 MJ/kg [23]), the energy recovered from their combustion is high enough to satisfy almost all the energy requirements of the process, with only a little natural gas contribution.

In general, the cast scraps have a better yield in r-MMA, with a lower production of waste, and they require a proportionally higher natural gas consumption compared to the mixed PMMA scraps (Table 1).

The quality of the feedstock affects the energy requirement of the process. High-quality waste, such as clean cast PMMA scraps, are more expensive, require less energy for the purification, and produce less by-products. Regarding the net energy recovery, cast scraps will require slightly more external energy source than mixed PMMA waste scraps. Although the MMAtwo consortium alone has committed to collect and treat up to 27 000 tons/y of PMMA by 2023, the current European plant size is around 5000 tons/y, as the experience of Madreperla, Monomeros del Valles, and in the past Evonik, Arkema, or Du Vergier [9,24] suggests. While sorting and collection volumes ramp up, especially for end-of-life waste, highly industrialized and densely populated areas can easily have access to similar volumes. Therefore, we believe that 5000 tons/year of scraps is what any new plant should realistically aim to start with, especially while establishing a sound market.

### 2.2. PMMA—Our Case Studies

Since the nature of scraps affects the mass and energy balance of the process, to investigate the PMMA recycling options, we benchmarked cast and mixed scraps, on a common final r-MMA purity. In general, for any given r-MMA grade, mixed scraps present a lower global (depolymerization + purification) r-MMA yield, compared to cast (78 wt.% vs 90 wt.% respectively—Table 1). At the same time, mixed scraps produce proportionally more waste, but require less external source of energy. In Table 1, we report the mass balance for both cast and mixed scraps processes.

### 2.3. Economic Analysis

When a portfolio of potential projects is available, companies or public investors need to evaluate the potential profitability of each project, to assess when and if to take a risk. The profitability of a project can be assessed by different methodologies, such as the payout period plus interest (POP), the net present value (NPV), the discounted cash flow rate of return (DCFROR), and the uniform annual cost (UAC), among others [25]. The NPV, also known as net present worth (NPW), is the most common quantitative methodology to assess the profitability, since it considers the effect of the time value of money on the profitability [25], and the annual variation in expenses and revenues [26]. The NPV of a project is the sum of the present values of future cash flows [26]:(1)$$\mathrm{NPV}=\overset{n=t}{\underset{n=1}{\sum }}\frac{CF_{n}}{{\left(1+i\right)}^{n}}$$
where *CF_n_* represents the annual operational cash flow for year *n*, *i* is the interest rate, and t the project life in years.

When the NPV is positive, the project earns more than the interest (discount) rate selected, or the best alternative project [25]. The annual operational cash flow *CF_n_* is the sum of the annual net profit after taxes (ANP), and the depreciation D:(2)$$CF_{n}=\mathrm{ANP}+D$$

With ANP being the annual gross profit—the taxes (TR). The annual gross profit (AP), is the sum of sales S, minus expenses E, minus depreciation D, so that the annual operational cash flow *CF_n_* becomes:(3)$$CF_{n}=\left(S-E-D\right)*\left(1-\mathrm{TR}\right)+D$$

The sales are the revenue generated from the r-MMA, the expenses are the total expenses of the plant, the tax rate is set at 35% as for France, and the depreciation follows the straight line depreciation on 10 years [25]. In our assessment, we divided the expenses in *Fixed* and *Variable*. The *Fixed* expenses are:Direct labor;Operating supervision, assumed as 18% of the direct labor;Laboratory charges, assumed as 18% of the direct labor:Plant overhead, assumed as 60% of direct labor;Administration, assumed as 20% of direct laborMaintenance and repairs, assumed as 2% of the total investment, but which could be higher for a cheap equipment;Operating supplies, assumed as 1% of the total investment;Financial interests, assumed as 2% of the total investment;Property taxes, assumed as 2% of the total investment; andInsurance, assumed as 2% of the total investment.

The *Variable* expenses are:Raw Materials (PMMA scraps, natural gas);Waste disposal (Solid waste, heavy ends);Other utilities (water, electricity, etc.) assumed as 3% of total raw material and waste disposal because we did not go into that level of details;Distribution and selling, as 5% of the sales, because we assumed here to have a limited pool of customers, that therefore minimizes the distribution related costs;R&D and Royalties, as 3% of the sales in total. This way R&D budget depends on the sales, but remains in the range of what is seen for commodity chemicals.We also made the additional general assumptions for the plant:Plant life is 20 years;The plant is located in France;Operating time is 8000 h, or 330 days;Depreciation is in 10 years;Internal rate of return (IRR) is 10%;Labor cost is calculated for 5 shifts per day, continuous process;The plant needs 2 operators per shift, at a cost of 60 000 USD/y per operator;The capital investment is spread over two years (year 0 and year 1), where 2/3 of the capital is invested in the first year, and 1/3 in the second;The plant starts to operate in the third year, at 50% production capacity, then it ramps up to 75% in the fourth year, 90% in the fifth, and finally 100% starting from the sixth year (year 5 in following tables and figures).

At this early stage, the capital investment can be estimated in different ways.

In 2017 Tsagkari et al. reviewed cost estimation methods for biomass conversion processes at class 4–5 AACE (Association for the Advancement of Cost Engineering) [27,28], pointing out that there are several methods for early stage estimation. In our case, we can rely on the Petley [29], and Lange methods [30]. The Petley method estimates the ISBL of the plant starting from the number of functional units, the capacity of the plant, the maximum pressure and temperature. In 2001, Lange proposed two capital cost estimation methods, one based on the plant energy losses, and one on the plant energy transfer. Both methods are appropriate for highly exothermic, or endothermic processes and start to be reliable for energy transfer/losses higher than 10 MW.

The original Petley’s correlation (from 1988) updated to 2019, and relocated to France is:(4)$$ISBL\;\left(2019\right)=55882Q^{0.44}N^{0.486}{T_{\max}}^{0.038}{P_{\max}}^{-0.22}{F_{m}}^{0.341}\frac{\mathrm{CEPCI}\left(2019\right)}{\mathrm{CEPCI}\left(1988\right)}\cdot Fl$$
where *ISBL* is the Inside Battery Limit investment, *Q* is the plant capacity in tons/y of scraps, *N* the number of process steps (4 here), *T*_max_ the maximum process temperature in K (723 K), *P*_max_ the maximum pressure in bar (1.5 bars), and *F*_m_ the material construction factor (1.5). The Chemical Engineering Plant Cost Index (CEPCI) updates the plant cost from 1988 to 2019, and the relocation factor (*Fl*) relocate it to France. In 1988 the CEPCI was 342.5, while in 2019 it was 607.5 [31,32]. The Peters *and* Timmerhaus handbook suggests that the Outside Battery Limit (OSBL) investment, is 25–40% of the ISBL [33], so we selected 40% to be conservative.

Lange correlated the ISBL with the energy losses in the plant, calculated as LHV_(feed+fuel)_ − LHV_(product)_:(5)$$ISBL+OSBL\;\left(2019\right)=3.0\cdot {\left(\mathrm{energy}\;\mathrm{losses}\;\left[\mathrm{MW}\right]\right)}^{0.84}\cdot \frac{\mathrm{CEPCI}\left(2019\right)}{\mathrm{CEPCI}\left(1993\right)}\cdot Fl$$

Again, we updated the investment with the CEPCI of 1993 (343.5), and the relocation factor for France.

Alternatively, he correlated the ISBL with the energy transferred in within the process, as:(6)$$ISBL\left(2019\right)=2.9\cdot {\left(\mathrm{energy}\;\mathrm{transfer}\;\left[\mathrm{MW}\right]\right)}^{0.55}\cdot \frac{\mathrm{CEPCI}\left(2019\right)}{\mathrm{CEPCI}\left(1993\right)}\cdot Fl$$

A fourth method is by expert judgment adjusted on existing plant capital costs. We know that the PMMA recycling process does not differ that much from a plastic to oil pyrolysis plant, some pyrolysis-based plastic to fuel, or some existing plastic to plastic recycling plants. In the last 10 years, several plants suitable for comparison have been built, in Europe or US, and even more have been announced in the near future (Table 2). Reviewing such plants, along with process reviews from expert companies (e.g., IHS [34], or Nexant [35]), we gave a best educated guess on the capital cost of the plant.

In Figure 3, we report the capital cost versus capacity of some selected projects scaled to 2019 in France, as well as the estimated capital cost. For a 5000 tons/y plant, the two Lange methods reach their validity limit, (under)estimating a capex of 3.3 M USD, and 7.2 M USD respectively. The Petley method probably overestimates the capex at 18 M USD. Both methods are indeed “Class V” for process engineers, or technology readiness level “TRL 3–4” for chemists, and they carry significant uncertainties. For comparison, for similar plant sizes, announced investments for recycling plant were between 11 M USD, and 25 M USD for the first of a kind (FOAK) Plastic energy Plastic to Oil plant [36]. Obviously, FOAK plant are more expensive than n^th^ of a kind (NOAK) for the same technology/company, because of learning curve and technology maturity [37]. For instance, this is the case of Plastic Energy, Quantafuel, or QM Recycled Energy, in Table 2.

To select the most appropriate investment at this early stage, we extrapolated the data of (Figure 3) with the power law method, scaling everything to 5000 tons/y:(7)$$\frac{C_{1}}{C_{2}}={\left(\frac{S_{1}}{S_{2}}\right)}^{0.65}$$
where C is the Cost (ISBL or OSBL, etc.) and S the plant size, the scaled 5000 tons/y would therefore be in the 6 to 16.5 M USD range. On the basis of different evaluation methods, we assumed a reference capital investment of 7.5 M USD, that translates into a median (p50) CAPEX of 10 M USD (p10 at −20% reference Capex (i.e., 6 M USD); and p90 at +120% (i.e., 16.5 M USD)—Figure 3), and which also includes working capital and start-up costs. When a plant starts, it always needs raw material/product stocks, and some budget allocated for unforeseen contingencies.

### 2.4. Monte-Carlo Simulation

To factor-in all the variables uncertainties at once, as opposed to inspect only one variable at the time (sensitivity analysis), we performed a probability risk analysis, also known as Monte-Carlo simulations. The way we described it so far, the NPV method is a deterministic method, meaning that we obtain a single output when plugging in a fixed input. However, if we input probabilistic distributions instead, we obtain an output with its own probabilistic distribution. In the Monte-Carlo simulation technique, each input has its own statistical distribution (normal, log-normal, gaussian, gamma, etc.,), from which the calculator selects one random value. From the random input series, the simulation calculates the output variable, for instance the NPV. The process is then repeated a high enough number of times so that the output becomes reliable, for instance in the order of thousands. Eventually, out of the frequency of recurring of each outcome, we can calculate the probability of happening of a certain outcome (e.g., NPV on a certain year). The Perry’s Handbook, or our previous paper on Risk assessment using Monte-Carlo simulation is a good starting reference on this technique [25,49,50].

In layman’s terms, the risk assessment analysis consists in:Based on production price index-adjusted historical data, set a statistical distribution for the future price of raw materials (PMMA, natural gas), waste (solid wastes and heavy ends), and product (r-MMA);Based on the combination of cost estimation techniques, plant historical investments, future announced investments, and expert judgement, set a statistical distribution for the capital cost;Based on the inflation adjusted historical prices, find the historical correlation between raw materials, waste, and product price;Based on the expert judgment, give the best educated guess on the raw materials, waste, and product future prices correlation;For each input (capital cost, raw materials, waste, and product prices) adjust the random variables for the correlation matrix. In this way, the inputs do not vary independently, but we force their variation according to the correlation matrix;Determine the profitability (i.e., NPV, production cost);Repeat the last steps 3000 times;Evaluate the probability of each occurrence to happen, and look at the median probability, the p10 and p90 (10 % and 90 % probability) for the variables of interest.

The Monte-Carlo simulation copes with the uncertainties on raw material price, waste price, product price, and investment. This approach is very appropriate, as a risk analysis, in this very early stage evaluation of processes, to identify the most viable scenario leading to a successful project. We kept all the other values as a fixed deterministic variable (labor cost, other utilities cost, depreciation, taxes, all the other fixed costs).

#### 2.4.1. Raw Materials

The main raw materials are the PMMA scraps, and the natural gas to compensate for the extra heat needed.

The MMAtwo consortium estimates that before the Chinese trade ban, Europe was exporting between 15 000 tons/y and 30 000 tons/y to India and China [10], with around 8 000 traded within Europe [9,10]. Hence, we collected some market prices for PMMA scraps for the 2013–2016 period in the major ports of India (Figure 4) from Zauba [51], and Eximpulse [52]. What we see here are relatively good quality scraps, most likely cast materials, traded as “Acrylic scraps HS39159030”. Lower quality scraps are difficult to trace, also because they might be sourced from the local market. For this reason, we assumed two different statistical distributions for the PMMA scraps (Figure 5):The **“Cast” PMMA scraps**, more expensive because they are a higher purity kind of scrap; andThe **“Mixed” PMMA scraps**, a lower quality, cheaper feedstock.

In both cases, we assumed a log-normal distribution for the future price of the scraps. The “Cast” scraps have a {p10, p90} interval in between {305, 590} USD/t, while for the “Mixed” we selected {100, 220} USD/t. There is also a **third**, extreme **scenario**, where the scraps quality is so low, that they become available at zero price.

For the European Natural gas, we collected the monthly price in USD between 2000 and 2020 from FRED (Federal Reserve Economic Data) [53] (Figure 6). Then, we adjusted for inflation, and we found out that the prices can be fitted best with a log-normal distribution.

We do not see any reason why natural gas should change its statistical distribution over time in the near future, nor we think that its value should deviate much from last years’ trend. Therefore, we assumed a log-normal distribution for the natural gas price, with a {p10, p90} interval of {190, 705} USD/t, with the mean lying around 300 USD/t (Figure 7).

#### 2.4.2. Products—r-MMA

The recycled MMA (r-MMA) price depends strongly on the product quality. PMMA producers aim at the very best monomer quality that satisfies their needs, which could be up to that of the virgin material (>99.8% wt. purity [54]). At the same time, the western world, and in particularly Europe, is pushing toward sustainable development, and promoting circular economy models. For instance, Europe recently updated its circular economy action plan (March 2020 [55]), toughening the policy on waste export outside Europe, and it imposed a contribution of about 1000 USD/t for non-recycled plastic packaging to curb plastic waste [56]. In this framework, we foresee how recycled MMA will play an even bigger role hereafter. We collected the virgin MMA spot historical prices in EU (Figure 8), and we adjusted them for inflation with the PPI (producers price indices) index. The PPI index, available online on the organization for economic co-operation and development (OECD), is an “advanced indicator of price changes throughout the economy” that measures the average price change over time in selling price from the sellers’ perspective [57]. In this way, the only factor playing a role in the historical prices’ trends is geo/political.

Virgin MMA price is highly correlated to crude oil price (Figure 8), and we believe that this will not change in the near future.

We also collected some prices for exported “regenerated black MMA” HS 29161400 in the ports of India between 2013 and 2016 [61] (Figure 9), a period during which crude oil prices varied a lot. All shipments were destined to Asia (e.g., China, Bangladesh, or Pakistan), for an average tonnage of 400 ton/y. In the period of interest, this lower quality crude MMA product was selling at an average of 30% discount over the virgin material.

Again, the best fit for the PPI adjusted historical prices of virgin MMA was the log-normal distribution. For our future prediction, we imagined different r-MMA prices, for three different product’s qualities, each with a log-normal distribution (Figure 10):**Virgin-like MMA** price;**20% discount MMA** price. In western markets it will be very unlikely to see very low-quality products. Requests for r-MMA will increase due to stricter policies and increasing environmental awareness of the final customers. Therefore, we expect that recycling companies and PMMA producers will settle midway for a fairly good enough “low-quality” r-MMA;**20% premium MMA price**. Either for increasing requests, superior quality, or market fluctuations, we believe that a scenario where PMMA (or other MMA derivatives) producers will pay a premium for their r-MMA is plausible. It could also correspond to subsidies given to recycled products based on their quality.

#### 2.4.3. Waste—Heavy Ends and Solid Waste

The depolymerization step produces char and dust as cracking by-product, while the purification train separates the heavy ends from the recycled monomer. Carbonaceous waste from the depolymerization is collected and either sold as charcoal to heavy industries (e.g., steel), or landfilled. Heavy ends can be either burnt on site for heat recovery, or if their quality as combustible is too low, they must be disposed of. In our case, we assume to remove the very heavy ends in the evaporator V-3001 (Figure 1), in form of a syrup-like fluid. Then, the distillation fixes the purification and removes some slightly lighter heavy cuts that are burnt on site for energy. In the mass balance (Table 1), we assumed the very heavy ends to leave the plant as by-product. In the economic assessment, this oil could either be sold as a heating oil equivalent or disposed of. To be conservative, we assumed that there is no market for these very heavy ends, and they must be disposed of.

Same applies for the solid wastes. Because of the increasing public awareness, landfill taxes and gate fees have been swiftly increased in Europe in the last 20 years, as shown for instance by ADEME (French government agency) in a comparative report of 2017 [62]. As a reference, Finland had a landfill tax of 25 USD/ton in 2003 [63], which rose to 85 USD/ton in 2020 [64]. As per 2020, 23 EU members have a landfill tax, that varies from 10 (Latvia) to 150 USD/ton (Belgium) [65], as well as Switzerland and UK (150 USD/ton) [64].

To this tax, we still need to add the gate (or tipping) fee, at the entrance of the landfill. Gate fees depends again on the country, but they roughly are comparable to landfill taxes [64]. We believe that in the near future, landfill taxes will keep increasing in EU, leaning toward the highest rates in the continent (i.e., central and northern Europe countries such as Belgium, Luxemburg, Denmark, Sweden, etc.,).

For both heavy ends and solid wastes, we assumed a gaussian distribution. For solid wastes we assumed a median cost (landfill tax + gate fee) of 250 USD/t and a deviation of 50 USD/t, and for the heavy ends a median cost of 150 USD/t (Figure 11) and a deviation of 50 USD/t. In this way, we assume to treat the plant by-products according to the highest European standards in terms of waste disposal, as opposed of what happens in existing lower quality PMMA recycling plant in developing countries (e.g., India, Brazil, etc.,).

#### 2.4.4. Investment

In a study of 2012, J.K. Hollmann queried over 1000 projects in different industries, and demonstrated how the p10/p90 accuracy range of the capital investment, is close to −20%/+120%, in a log-normal statistical fashion [66]. Thus, in our Monte Carlo simulation, we assumed a log-normal distribution for the capital investment, where p10 corresponds to −20% of the capital cost, and p90 to the + 120%. 

For the analysis, we selected four conditions:**Full CAPEX at 10 M USD** (Figure 12), for a higher capital-intensive plant. This is the case when we have lower quality scraps (MIXED or 0 price scraps);**30% subsidies on Full CAPEX** for a final CAPEX of **7 M USD** (Figure 13). It is again a higher capital-intensive plant treating lower quality scraps. Local authorities are more willing to provide subsidies when they see the benefit of the money they spent. In this case, lower quality scraps might need higher cost at pre-treatment or purification, and subsidies could cover that;**Full CAPEX at 7 M USD** (Figure 13), for a slightly lower capital-intensive plant. We imagine this to be the case when treating only Cast scraps. The feedstock is relatively “clean”, and a cheaper plant is expected; and**Full Capex at 1.3 M USD.** This is the case of a cheap technology, or a plant made of second-hand equipment. In both cases, the lower capital expenditure is likely to be accompanied by a shorter plant life (e.g., 10 years instead of 20), an overall lower quality of the final product, and so a lower price.

## 3. Results

### 3.1. Scenarios

To study the challenges and possibilities of the recycling industry at large, we compared the profitability for different scenarios (Table 3). The scenarios differ for kind of feedstock (Mixed or Cast), final product quality and selling price (20% discount, virgin or 20% premium price), or CAPEX. In Table 3, we summarize the scenarios, and the prices distribution for raw materials, waste/by-products, and products. We ended up outlining nine different scenarios (Table 3):**Scenario 1**—The plant treats mixed scraps, with a full Capex at 10 M USD, and the r-MMA sells at virgin price. This is our base case. Processing mixed scraps needs a relatively higher capital-intensive plant to reach an acceptable purity for the EU market.**Scenario 2**—Like Scenario 1, but the project obtains external funding as grants or subsidies (30% is an accessible value for an European plant);**Scenario 3**—Like Scenario 1, but the r-MMA purity requested is higher, and it sells with a premium of 20%. This is more representative of what could happen in the long-term;**Scenarios 4** and **8**—The plant treats mixed scraps, with a second-hand, or a cheap process technology (e.g., dry distillation). Therefore, the r-MMA produced is of a lower quality and sells at 20% discount compared to the virgin. In Scenario 4, the plant life is still 20 years, while in Scenario 8 the plant is replaced after 10 years.**Scenarios 5** and **6**—The plant processes Cast scraps, more expensive than the mixed, but requiring less/cheaper equipment. The r-MMA sells at virgin or premium price, in Scenarios 5 and 6, respectively;**Scenarios 7** and **9**—PMMA scraps are very low quality, and therefore available for free. However, the plant still needs a relatively high investment (10 M USD) to purify the MMA to a grade pure enough to at least enter the market. Nonetheless, r-MMA sells at a 20% discount price. This scenario is representative of very polluted end-of-life streams, where sorting is difficult due to poor infrastructure. For instance, it could apply to the automotive or the construction industry; Scenario 7 is at full capex (10 M USD), while Scenario 9 is with 30% subsidies. Because the project repurposes waste that would otherwise inevitably end in landfill, local bodies grant subsidies for environmental reasons rather than economic reasons only. Our guess is that for the most part, these two scenarios are probable to be transient. The public opinion calls for more sustainable materials, and the government agencies legislate accordingly. Hence, the collecting and sorting infrastructure will eventually keep up with the demands for scraps. When that will happen, PMMA producers in EU will not easily settle for a poor-quality monomer.

### 3.2. Correlation Matrix

In 2019, the OECD (Organization for Economic Co-operation and Development) demonstrates that, with few exceptions, waste generation “is still very much linked to economic growth” [67]. Therefore, although prices for feedstock and products/waste vary with time, such variations are not completely random. When the economy is booming, people get better salaries, eat more (or better food), and their purchasing power increases. The downside is that energy and food prices increase too, as well as waste production, especially plastic waste [68]. For instance, Germany has a GDP per capita of around 40 000 USD, and produces 0.5 kg of plastic waste per capita, same applies for the Netherlands (Figure 14). On the contrary, Albania has a GDP per capita of 10 000 USD, and produces less than 0.1 kg plastic per person (Figure 14).

Moreover, in our case, PMMA scraps price is highly correlated to r-MMA price, and r-MMA price is correlated to price of energy (Natural Gas for us). Same applies to the price of waste disposal that are expected to be linked. To be conservative, we decided to consider the heavy ends as a waste. However, we know that its properties are like those of a heating oil [71], so historically it has been correlated to natural gas, as well as MMA. For PMMA scraps very scarce historical data are available, or in a very narrow timespan (e.g., Zauba imports 2013–2017). Based on the historical data (if any), and our expert judgment, we represented our vision of the future in a correlation matrix, that mimic prices trends fluctuations at the best of our knowledge (Table 4). Once correlated, the prices for the 3000 Monte-Carlo cases appear as in Figure 15.

### 3.3. Profitability Analysis by Monte Carlo Simulation

Under our assumptions, the base case (Scenario 1), where the plant is built without subsidies, treating 5000 tons/y of mixed scraps to produce 3889 tons/y of r-MMA selling at virgin price is not very appealing. Over 3000 simulations, the project has a (median) payback time of 8 years (Figure 16) with 30% probability to make no money after 10 years of production (Figure 17). To understand where assumptions improvements would be most effective, we studied the impact of the individual uncertainties of feedstock, waste, product, and investment, on the median NPV. In Scenario 1, when the parameters vary between their 10% and 90% probability of their statistical distribution, the NPV changes accordingly to the impact that the single uncertainty has on it. The r-MMA price has the largest influence on the NPV, comparable to that of the investment cost (Figure 18). This means that improving the assumptions on the statistical distribution of either, should improve the uncertainties on the NPV, and lower the risk of losing money. For instance, if the investment uncertainties reduce as low as the equivalent of a Class 3 AACE (extreme values of the tornado at p29 and p75), the investment has an impact on the NPV comparable to that of PMMA scraps (Figure 19). To reach the confidence of a AACE class 3, we need a well identified list of equipment. After class 3, the improvement is marginal (at this stage) (Figure 19). In the Tornado graph (Figure 18), the investment varies in between −20% and +120% (p10, p90), of the selected investment (7.5 M USD), whose median is 10 M USD. Instead, the tornado capex plot (Figure 19) varies from the median. For instance, Class 5 in Figure 19 corresponds to a [p10, p90] of −50% and + 100% of the median, while the [p10, p90] in Figure 18 are the 10th and 90th percentile of that log-normal distribution having 10 M USD as median. The Tornado graph was generated to illustrate the impact of the definition of the project (Capex) on the dispersion of the NPV values.

When Scenario 1 gets either 30% subsidies on the Capex (Scenario 2), or sells r-MMA at 20% premium price (Scenario 3), the project becomes much more viable (Figure 16). Both solutions reduce the median payback period to 5/6 years; while the 20% premium solution (Scenario 3) becomes more profitable on the long-term (Figure 16). Both Scenarios (2 and 3) are equally attractive, but they still have a 10–15% probability of not earning any money after 10 years of production (i.e., probability at 0 NPV) (Figure 17).

In the current political climate, we cannot really give an excessive premium on the r-MMA price, but we do have a good case to present to local authorities when asking for subsidies. Scenarios 4 and 8 are representative of a second-hand plant, or a very cheap technology (Capex 1.3 M USD), that treats mixed scraps to produce low-quality r-MMA selling at 20% discount price. When operating a low capital-intensive plant, provided that the r-MMA quality is still acceptable to enter the market, we cover the plant cost after a little more than 1 year of production (Figure 20). The fast payback time makes this a low-risk alternative. However, with time, the higher r-MMA selling price overcomes the effect that a low depreciation has on the expenses. At the end of the plant life Scenario 2 has the same NPV as Scenario 4. Moreover, if we expect to replace the second-hand equipment/low-quality plant after 10 years, eventually Scenario 8 has a lower NPV than Scenario 2, almost comparable to that of Scenario 1. With the difference that while Scenario 1 has still room for improvement (subsidies, higher r-MMA price due to EU policies), Scenario 8 has limited growing potential. This suggests that public or private investors should look at solid technologies, which are able to provide a good quality product, rather than very cheap plants, even if the former are more expensive up-front.

When comparing Cast and Mixed scraps, Cast scraps makes for an overall better case. Cast scraps are more expensive, but they can be used in a cheaper plant, and have less carbon losses along the process, making for a better r-MMA global yield. We compared the base case (Scenario 1), with the case of a 5000 tons/y cast scrap 7 M USD plant, producing 4512 tons/y of r-MMA selling at virgin price (Scenario 5), or 20% premium price (Scenario 6) (Figure 21). At full investment cost, selling at virgin price, the mixed scrap plant (Scenario 1), falls short compared to its Cast equivalent (Scenario 5). Over the plant life, the cumulative NPV plot slope is the same in these two cases (Scenario 1 and 5). Namely, the better r-MMA cast global yield compensates for the higher feedstock price, with respect to the Mixed scrap case. Similarly, Scenario 5 and Scenario 2, i.e., the case where we get 30% subsidies on the mixed scrap plant, have NPVs which basically overlap. This means that the investment plays here a fundamental role. At parity of investment and selling price, the economic viability is not influenced by the kind of scraps. However, mixed scraps consist in post-production, but end-of life PMMA as well. Should PMMA producers be willing to pay a premium for r-MMA from end-of-life products only (the only ones that have made a full cycle in the economy), for instance to better advertise their product, the mixed plant would have a more positive case than the cast (Scenario 3 vs 5), even at a higher Capex.

The overall superiority of the cast over the mixed scenarios appears also on the NPV probability curve at the 10th year of production (Figure 22). Scenarios 5 and 6 have a probability of making zero money of 15% and 5% respectively, while Scenario 1 lost money around 30% of the times. Again, Scenarios 2 and 3 are equivalent to Scenario 5 in terms of cumulative NPV equal to zero at year 12 (10th year of production).

Nowadays, we see more and more techno-economic assessment on plastic recycling setting the feedstock price at zero [72], or even at a negative value [73]. PMMA is a relatively expensive plastic, compared to higher volume materials like PP, PVC, or PE. Therefore, as demonstrated by the few available historical import data, PMMA scraps have a market, nonetheless. However, the socio-political climate around the plastic recycling world is mercurial. Therefore, we wanted to investigate what would happen if our scraps were of such a low quality (e.g., end-of-life not completely sorted) to be available free of charge.

In this case, we can expect to either pay an extra for the pre-treatment (e.g., additional sorting, crushing, washing, and drying), lose in global yield to reach the target quality, or accept a lower quality monomer but with the same global r-MMA yield. We decided for this last option, and assessed the economics of a mixed scrap plant, treating unsorted scraps at 0 cost, selling low quality (e.g., 97–98 wt.% pure or less) r-MMA at 20% discount. We assumed such a plant to require the same capital expenditure of our base case (Scenario 1 at 10 M USD), and we evaluated it with (Scenario 9) and without (Scenario 7) subsidies (Figure 23). The raw materials contribution to the total expenses is now (Scenarios 7 and 9) much lower. In the global model, the other utilities are expressed in terms of a percentage of raw materials cost. In the case of the scrap at 0 price, with the mass yield of the mixed feedstock, we expressed the utility cost as a fixed value, equal to that of the base case.

Once more, the r-MMA quality differentiates between positive and negative cases. At first glance, scraps at zero cost may seem more attractive, but it is not correct. Scenario 7, the equivalent of Scenario 1, has a 40% probability of nullifying the NPV at the tenth year of production, as opposed to 30% in Scenario 1 (Figure 24). Besides, Scenario 7 cumulative NPV turns positive only after 9/10 years, 2 years later than Scenario 1 (Figure 23). The subsidized Scenarios 2 and 9, follow the same trend, with Scenario 2 being more profitable (Figure 23 and Figure 24). Even if, for environmental reasons, public bodies granted funds to the 0 scrap price plant only, this case (Scenario 9) gains the same profits (in median) than Scenario 1 at the end of the plant life, with the risk that a zero cost feedstock may see an increase of its price when alternatives develop (such as fuel users). To target high-quality product is just overall more viable.

Ideally, the best process technology can process all the scraps grade, yielding to the top quality r-MMA product, regardless the nature of the scrap itself. In the MMAtwo experience we found out how, on some extent, all the technologies tend to be feedstock sensitive (some more than others). These results suggest that any new player in the PMMA recycling field must demonstrate the maturity of their process, to produce high-quality r-MMA. Getting low quality scraps as cheap as possible, is not only difficult, but can be negative for the profitability of the plant. The only case where this could make sense, is if these scraps come with an important negative price.

### 3.4. Cost of Production

When the input variables are probabilistic functions, the cost of production is a probabilistic function as well. For all the cases, the cost of production before depreciation distribution fits well a normal, or log-normal distribution. For instance (Figure 25), Scenario 1 has a cost of production before depreciation between 784 and 1176 USD/ton for the interval of probability p5–p95 (green interval—Figure 25). Labor cost, the “other fixed”, and “other variable” costs are the most important contributors to the cost of production (Figure 26).

The “other fixed” costs comprehend (Section 2.2) maintenance and repairs, operating supplies, property taxes, financing interests, and insurance. The “other variable” costs are instead (Section 2.2) expressed as a % of the sales, includes R&D and Royalties, and distribution and selling.

We already assumed to run a lean plant, with only two operators per shift, which is the minimum for safety reasons. The only way to improve that, would be erecting the plant as a brownfield. In this way, we could run the plant with only 1 operator per shift, while there would still be somebody else on the industrial site.

The other fixed costs (as well as the depreciation) are linked to the investment. The only way to reduce that is to keep the plant layout as simple as possible. An idea could be to reduce the number of spare equipment, or minimize the side streams that have then to be treated on-site.

Classically, savings are made on the other variable cost, but not much can be done in our case. With only a total of 8% of the sales, there is a small room for improvement. Since the plant is in the middle of Europe, close to PMMA producers and scrap collectors, we already imagined keeping the distribution and selling costs quite low. Moreover, because there are numerous running technologies, we also assumed a low R&D and royalty contribution. Most of the processes have already demonstrated to be mature enough to produce sellable r-MMA. R&D should focus on tuning the right technology with the right feedstock available, or target product, rather than finding novel routes.

## 4. Discussion and Conclusions

To highlight challenges and opportunities of PMMA recycling we performed a Monte Carlo simulation with a total of 3000 iterations. The risk analysis debunks some preconceptions, and it is a support for any future work in the field.

Based on the results of our simulation, we ranked the Scenarios according to different selected outcomes: (i) Pay-back time; (ii) NPV after 10 y of operation; (iii) probability of losing money after 10 y of operation (Table 5).

The 5000 tons/y, 10 USD M plant of Mixed PMMA scraps, producing 3889 ton/y of r-MMA selling at virgin price, is the most representative case, and acts as baseline for comparison. We demonstrated how:A highly energy integrated process minimizes waste streams, utility consumptions, and environmental burden, as well as contributing to make a positive economic case;A median Capex of 10 M USD (or lower) is what companies should aim at for a 5000 tons/y PMMA depolymerization plant;NPV is most sensitive to the uncertainties on r-MMA price and investment. Expert judgment, experience on similar plants, subsidies, reduce the risk on the investment. Securing a deal with the r-MMA final buyers narrows down the r-MMA statistical distributions and improves the reliability of the business plan;New players should not compromise on product quality, even if this means higher capital plants, or “cleaner” and more expensive scraps. Whenever capital cost (Scenarios 4 and 8) and scrap composition (Scenarios 7 and 9) diminish the product quality, the overall economics worsens. Cheap plants allow for a faster pay-back time but are less profitable overall compared to the base case. The lower profits in the zero price waste scraps plant outweigh the reduced cost of operation.If we are in a market with a sufficiently high demand for low quality product (e.g., India, or Brazil), there is no need to over-purify the regenerated monomer, the plant is already economic as it is. However, the evolving EU legislation might hurdle the entrance of low quality r-MMA in the European market;Currently, Cast scraps plants are more viable than Mixed scrap plants, under the hypothesis that the first are less capital-intensive. However, when a plant is designed and built to treat both, it can switch with no effect on the cumulative NPV. The better global yield of Cast scraps counterbalances the lower feedstock price of Mixed scraps;The r-MMA cost of production follows a log-normal distribution, and labor cost, investment, and “other variable cost” are the main contributors.

Therefore, regardless the process technology, we outlined what are the minimum requirements that a plant should check in terms of mass balance, capital investment, feedstock and product prices, feedstock quality, and economic figures, in order to be competitive.

A first improvement for the model could be to investigate further waste streams. For instance, PMMA/fiber glass composites that might be employed in boats and wind blades, PMMA/ATH (aluminum trihydroxide), or solid surface materials, which are sold as a marble substitute. Future research is needed to understand how to best valorize each part of the composite, while still having a positive case.

An equally interesting scenario to investigate, would be when local governments grant subsidies on the product made, rather than the plant. This would be a variation of the 20% premium scenario we analyzed. Because of the volatile nature of the plastic recycling world, it would be wise for governmental agencies to subsidize only the winning horse, i.e., only when a company has already demonstrated production. At the beginning, the company takes all the risk, because they put money up-front. However, if the plant works, and it demonstrates production, subsidies on the product should guarantee a higher return than subsidies on the Capex.

Furthermore, r-MMA is currently sold in Europe under Reach registration exemption for recycled materials. In the framework of this study, it means that the regenerated monomer can directly enter the market. This might change in the future. As pointed out, r-MMA quality is feedstock sensitive and process sensitive, to some extent. Standards to select properly the PMMA scraps are then needed, in order to assign them the best value and address them to the different depolymerization technologies.

## Figures and Tables

**Figure 1 polymers-13-02724-f001:**
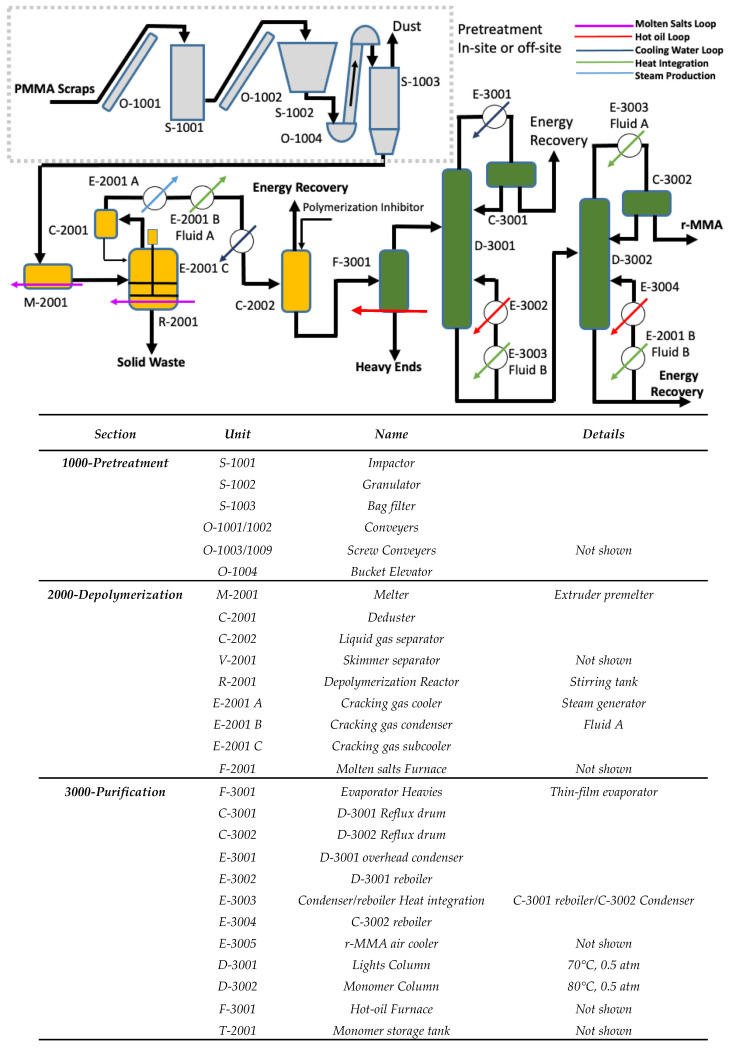
Simplified flowsheet for our design of a PMMA depolymerization process.

**Figure 2 polymers-13-02724-f002:**
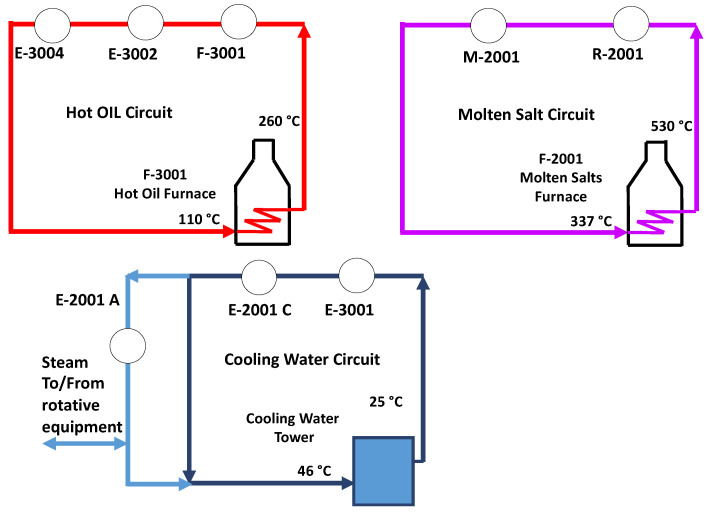
Utilities loop for a typical PMMA depolymerization process.

**Figure 3 polymers-13-02724-f003:**
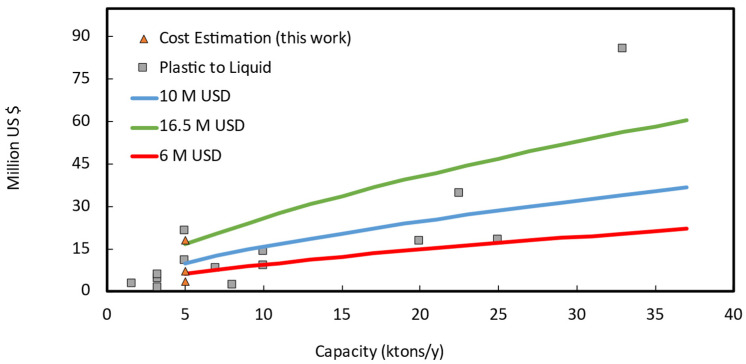
Scaled Investment (2019, France) versus original plant capacity. Historical data of plastic to liquid (oil or fuel) plants (grey squared marks), and early estimation methods (triangular marks) for a 5 kt/year scraps Mixed or Cast scraps PMMA depolymerization plant. The blue line represents the power-law with a reference investment of 10 M USD, while the green and the red with 16.5 M USD, and 6 M USD, respectively.

**Figure 4 polymers-13-02724-f004:**
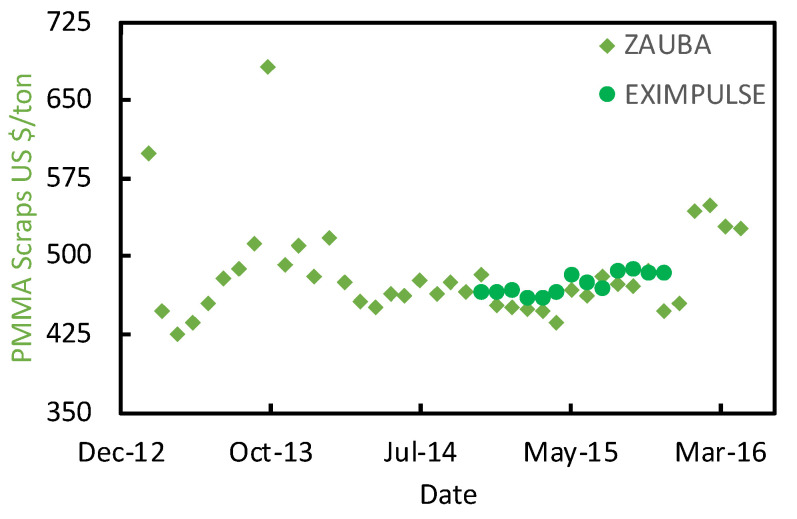
PMMA scraps import prices in India between 2013 and 2016, from Zauba [51], and Eximpulse [52].

**Figure 5 polymers-13-02724-f005:**
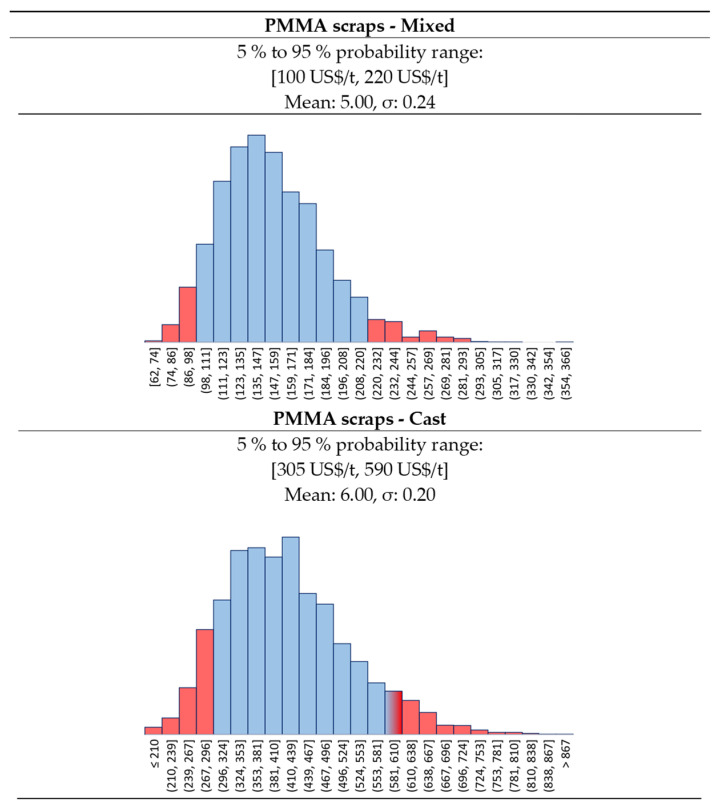
Log-Normal distribution for Mixed and Cast PMMA scraps prices.

**Figure 6 polymers-13-02724-f006:**
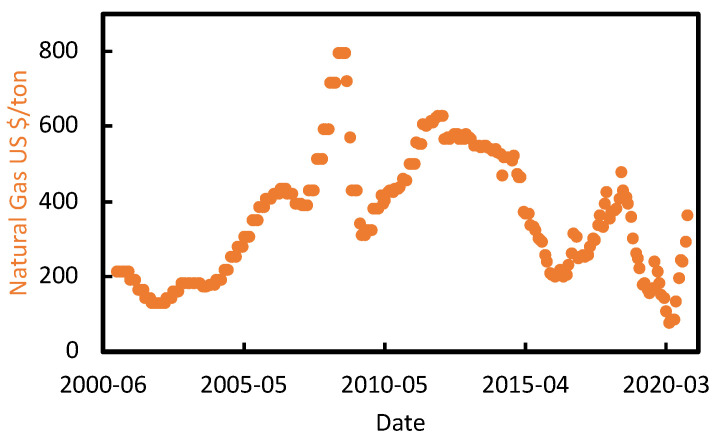
Historical natural gas prices in EU from [53].

**Figure 7 polymers-13-02724-f007:**
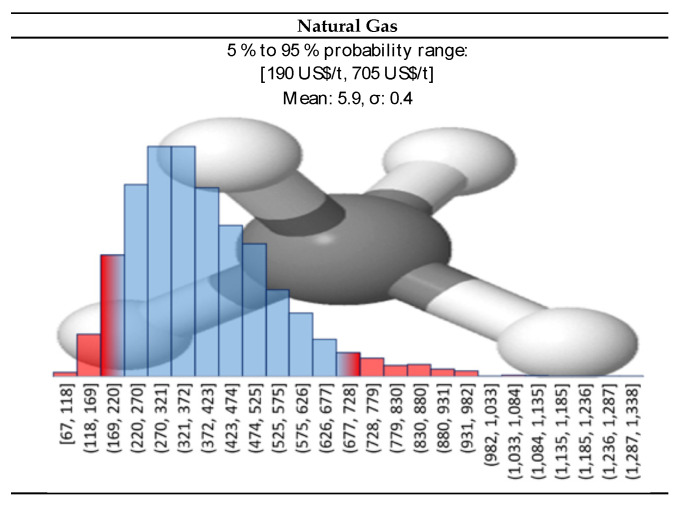
Log-normal distribution for natural gas prices.

**Figure 8 polymers-13-02724-f008:**
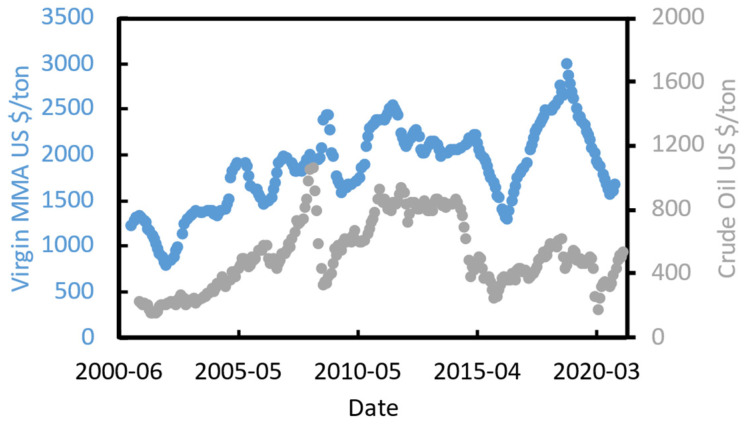
Virgin MMA spot prices from [8,58,59], and crude oil [60].

**Figure 9 polymers-13-02724-f009:**
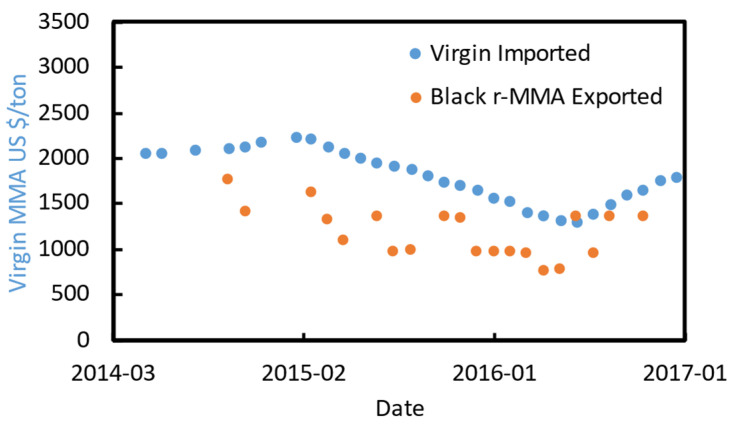
Crude exported MMA price [61], and virgin imported spot price between 2014 and 2017 in India.

**Figure 10 polymers-13-02724-f010:**
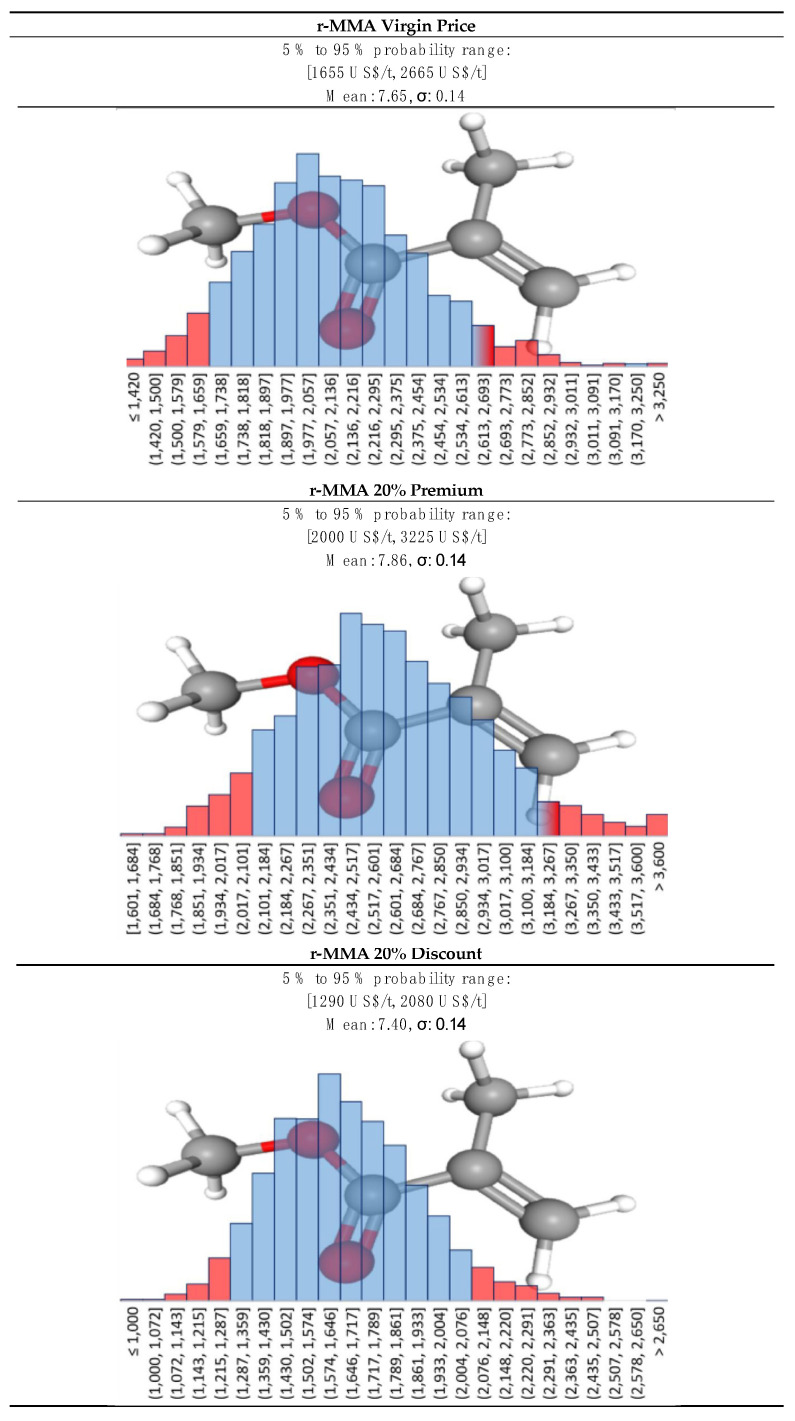
Log-normal distribution for r-MMA at virgin price, 20% premium, and 20% discount.

**Figure 11 polymers-13-02724-f011:**
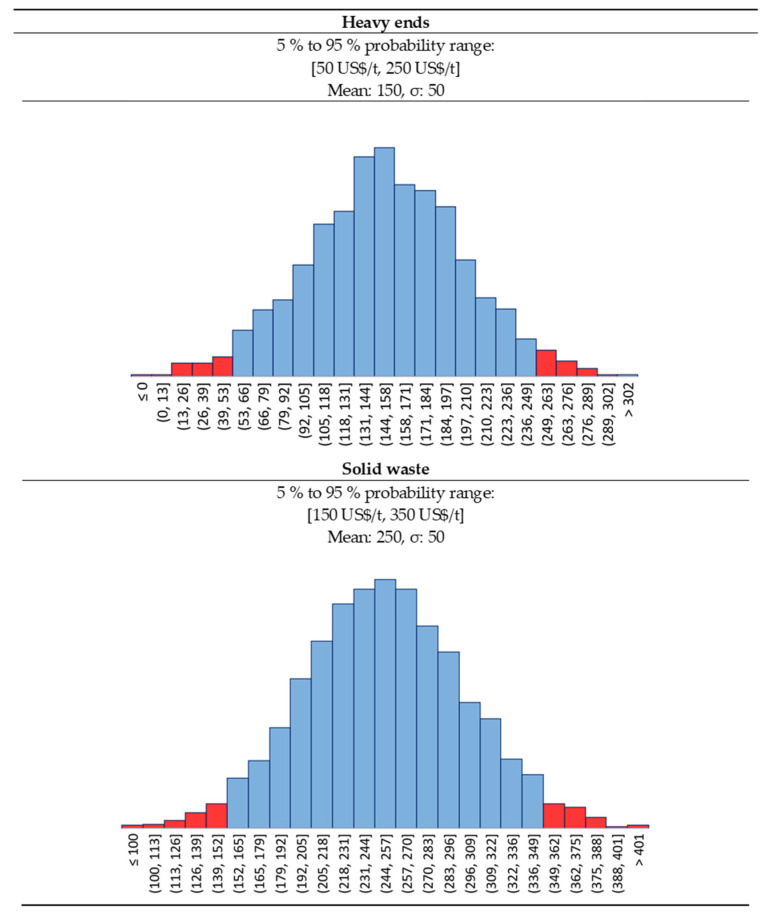
Solid waste and heavy ends statistical distribution.

**Figure 12 polymers-13-02724-f012:**
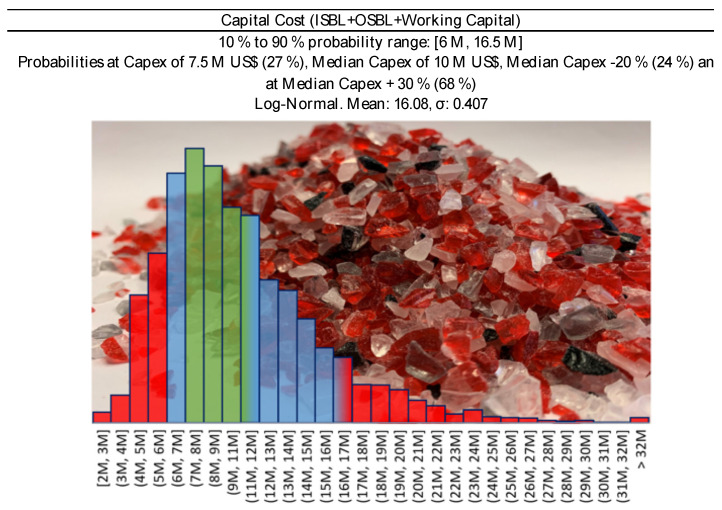
Capital cost distribution for 5000 tons/year PMMA scraps plant, at 10 M USD CAPEX—Capital Cost −20%/+ 30 % (Green Area) corresponds to Class 3 confidence interval of the Association for the Advancement of Cost Engineering (AACE) [28].

**Figure 13 polymers-13-02724-f013:**
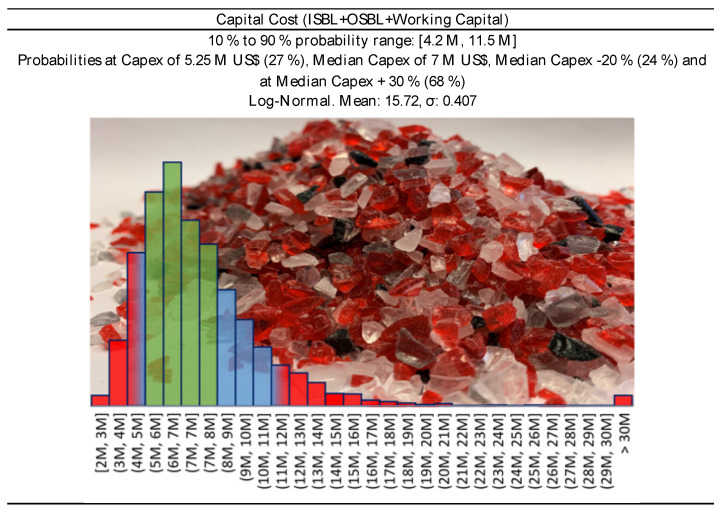
Capital cost distribution for 5000 tons/year PMMA scraps plant, at 7 M USD CAPEX—Capital cost −20%/+ 30 % (Green Area) corresponds to Class 3 confidence interval of the Association for the Advancement of Cost Engineering (AACE) [28].

**Figure 14 polymers-13-02724-f014:**
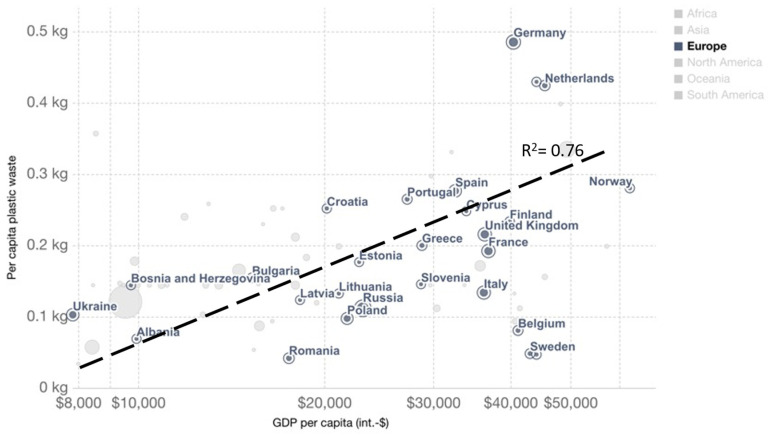
Per capita plastic waste vs. GDP per capita (2017), measured in 2011 international USD—From Our World in Data [68]—World Bank [69] and Jambeck et al. [70].

**Figure 15 polymers-13-02724-f015:**
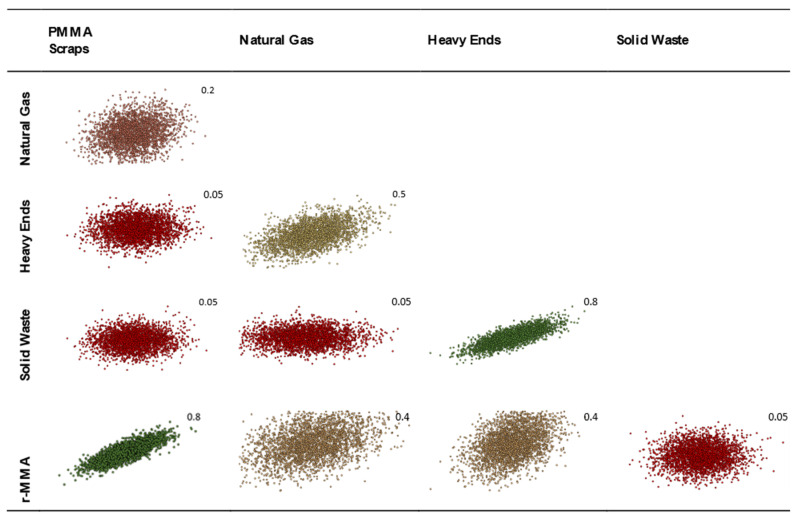
Visualization of the correlated prices of raw materials and products based on our vision of the future.

**Figure 16 polymers-13-02724-f016:**
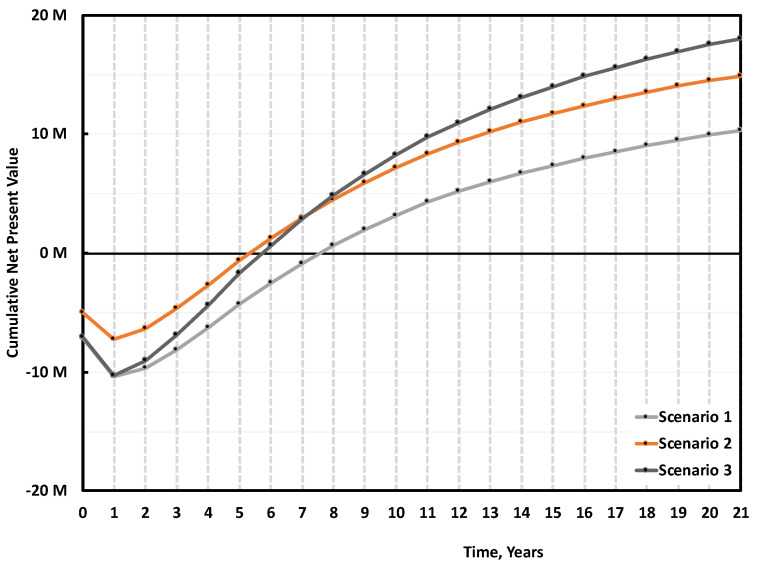
Median cumulative cash flow (NPV) over 20 years of production for 5000 tons/y plant, Scenario 1, 2, and 3 (Table 3). Scenario 1 is the base case, mixed scraps, Full 10 M USD Capex, and r-MMA sold at virgin price. Scenario 2 is when such a project gets 30% subsidies on the Capex, while Scenario 3 is when the r-MMA sells at 20% premium price.

**Figure 17 polymers-13-02724-f017:**
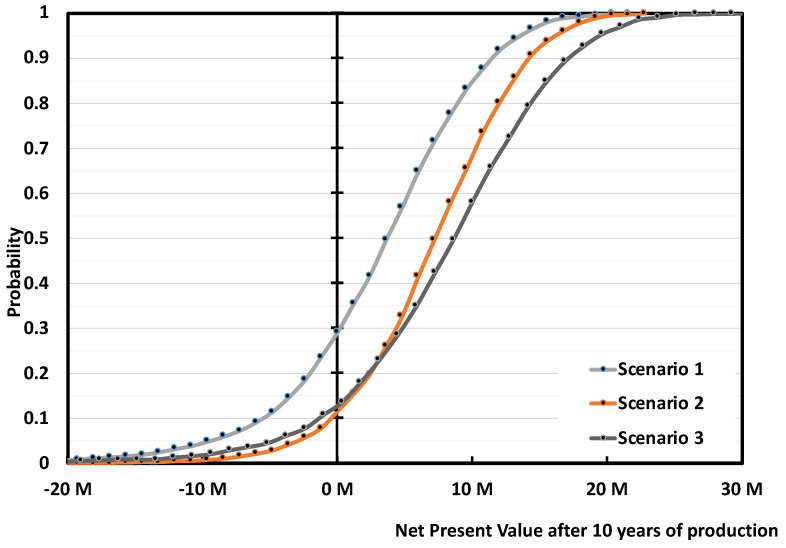
Probability vs Cumulative NPV over 10 years of production 5 000 tons/y plant. Scenario 1, 3 and 3. Scenario 1 is the base case, mixed scraps, Full 10 M USD Capex, and r-MMA sold at virgin price. Scenario 2 is when such a project gets 30% subsidies on the Capex, while Scenario 3 is when the r-MMA sells at 20% premium price.

**Figure 18 polymers-13-02724-f018:**
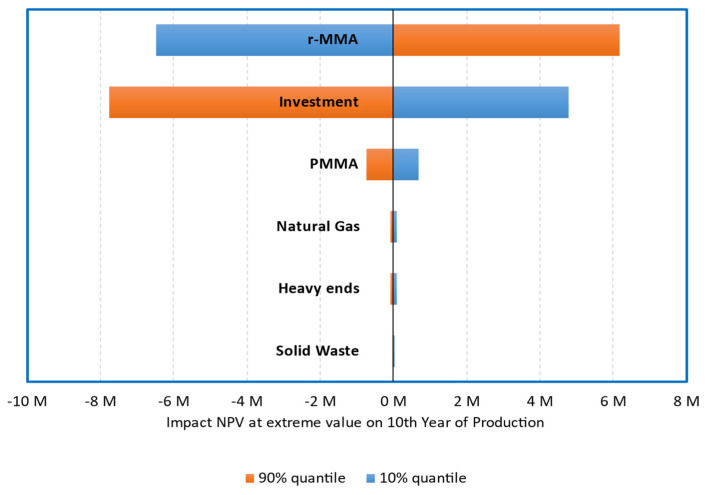
Tornado Plot for 5000 tons/y plant, **Scenario 1**. Effect of Raw-materials, waste, products, and investment variation (10% and 90% quantile), on the median NPV.

**Figure 19 polymers-13-02724-f019:**
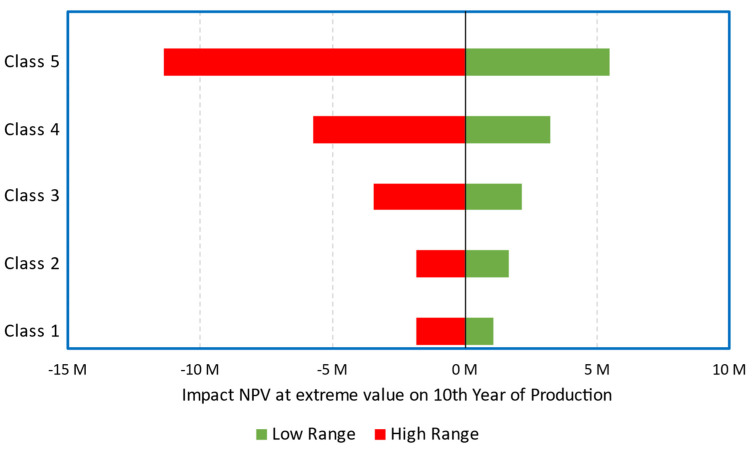
Scenario 1, impact of Capex class on the median NPV. Class 5 AACE corresponds to extremes at p4–p96 (or −50%, +100% of the median); Class 4 AACE corresponds to extremes at p85–p19 (or −30%, +50% of the median); Class 3 AACE corresponds to extremes at p29–p75 (or −20%, +30% of the median); Class 2 AACE corresponds to extremes at p34–p64 (or −15%, +15% of the median); Class 1 AACE corresponds to extremes at p40–p64 (or −10%, +15% of the median).

**Figure 20 polymers-13-02724-f020:**
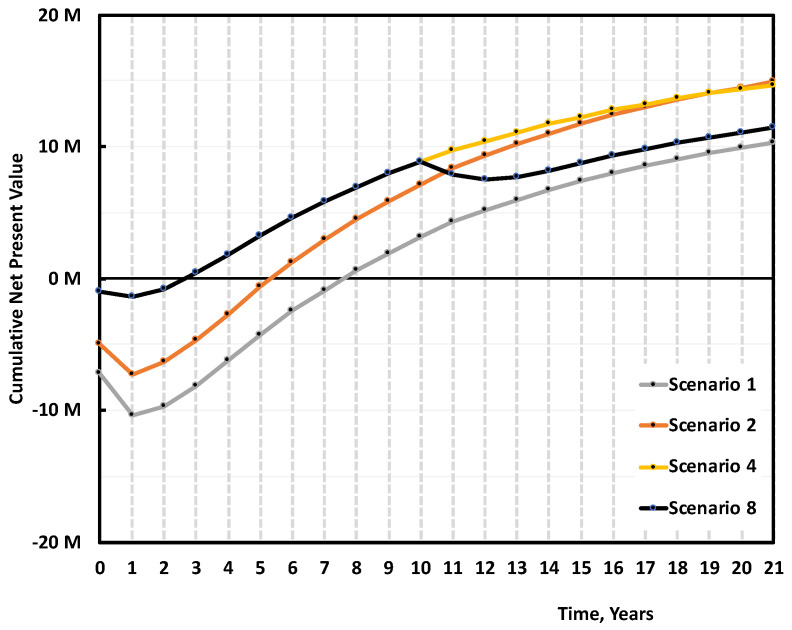
Median cumulative Cash Flow (NPV) over 20 years of production for 5000 tons/y plant, Scenario 1, 2 and 4, 8 (Table 3). Scenario 1 is the base case, mixed scraps, Full 10 M USD Capex, and r-MMA sold at virgin price. Scenario 2 is when such a project gets 30% subsidies on the Capex. Scenarios 4 and 8 are the case of a low-quality plant (1.3 M USD Capex), treating mixed scraps and selling r-MMA at 20% discount price. In Scenario 4 we operate the plant for the full 21 years, while in Scenario 8 we must replace it after 10 years.

**Figure 21 polymers-13-02724-f021:**
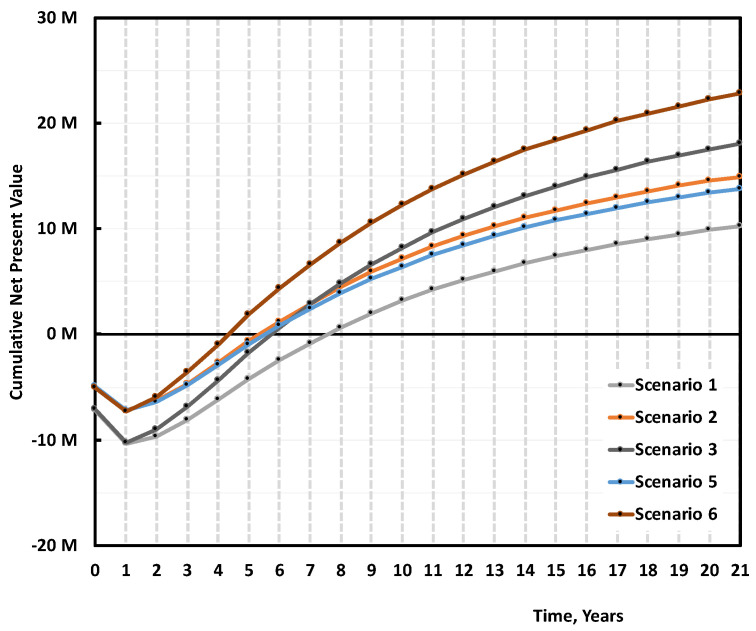
Median cumulative NPV over 20 years of production for 5000 tons/y plant, Scenario 1, 2, 3 and 5, 6 (Table 3). Scenario 1 is the base case, mixed scraps, Full 10 M USD Capex, and r-MMA sold at virgin price. Scenario 2 is when such a project gets 30% subsidies on the Capex, and Scenario 3 when r-MMA is sold at 20% premium price. Scenarios 5 and 6 are the case of a Cast scrap plant, that sells at either virgin (Scenario 5), or 20% price (Scenario 6).

**Figure 22 polymers-13-02724-f022:**
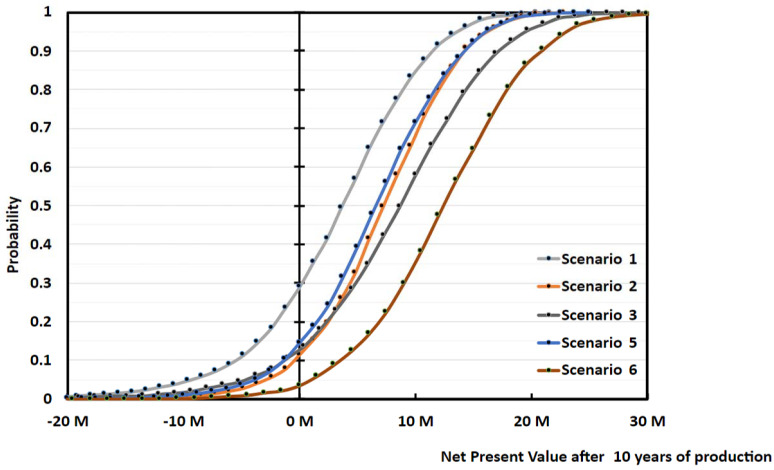
Probability vs Cumulative NPV over 10 years of production 5000 tons/y plant, Scenario 1, 2, 3 and 5, 6 (Table 3). Scenario 1 is the base case, mixed scraps, Full 10 M USD Capex, and r-MMA sold at virgin price. Scenario 2 is when such a project gets 30% subsidies on the Capex, and Scenario 3 when r-MMA is sold at 20% premium price. Scenarios 5 and 6 are the case of a Cast scrap plant, that sells at either virgin (Scenario 5), or 20% price (Scenario 6).

**Figure 23 polymers-13-02724-f023:**
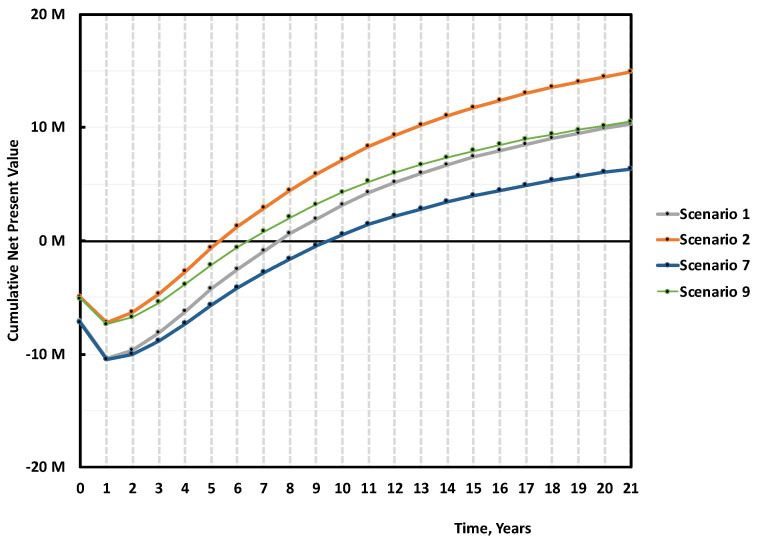
Median cumulative NPV over 20 years of production for 5000 tons/y plant, Scenario 1, 2, and 7, 9 (Table 3). Scenario 1 is the base case, mixed scraps, Full 10 M USD Capex, and r-MMA sold at virgin price. Scenario 2 is when such a project gets 30% subsidies on the Capex. Scenarios 7 and 9 are the case of a mixed scrap plant, with (Scenario 9) or without (Scenario 7) 30% subsidies, processing 0 cost low, quality PMMA, and selling r-MMA at 20% discount price.

**Figure 24 polymers-13-02724-f024:**
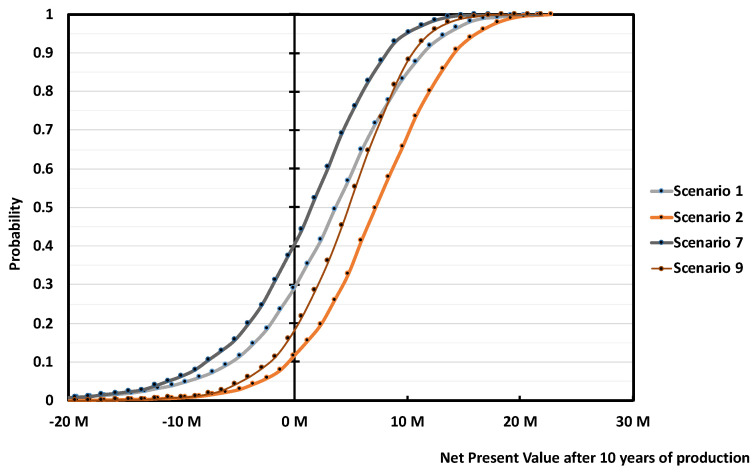
Probability vs Cumulative NPV over 10 years of production 5 000 tons/y plant, Scenario 1, 2, and 7, 9 (Table 3). Scenario 1 is the base case, mixed scraps, Full 10 M USD Capex, and r-MMA sold at virgin price. Scenario 2 is when such a project gets 30% subsidies on the Capex. Scenarios 7 and 9 are the case of a mixed scrap plant, with (Scenario 9) or without (Scenario 7) 30% subsidies, processing 0 cost, low quality PMMA, and selling r-MMA at 20% discount price.

**Figure 25 polymers-13-02724-f025:**
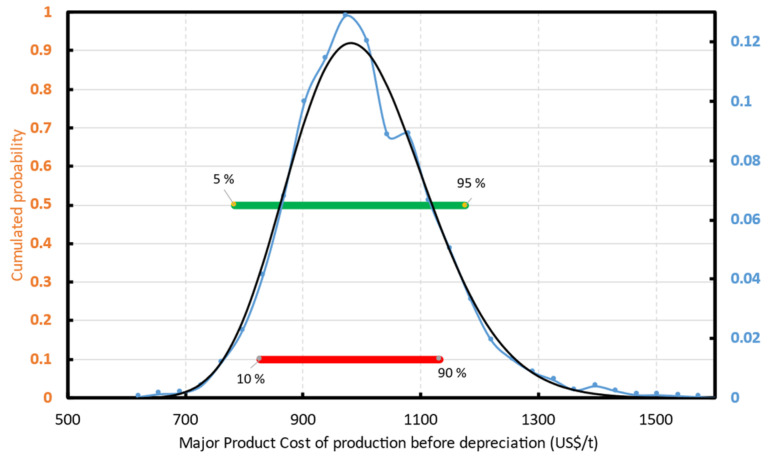
r-MMA cost of production before taxes and depreciation. Distribution for Scenario 1. The Blue line is the simulated distribution that fits a log-normal distribution (black line).

**Figure 26 polymers-13-02724-f026:**
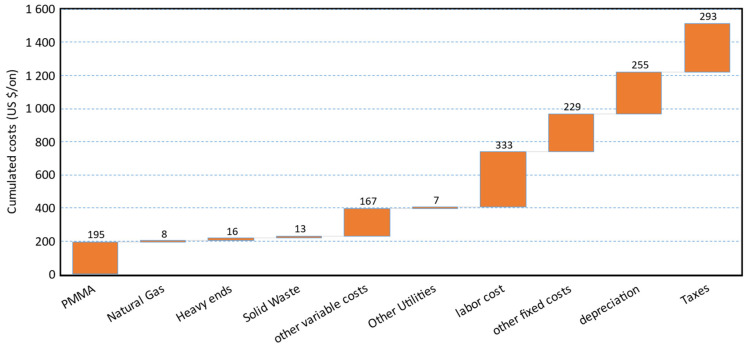
Median production cost cascade for Scenario 1.

**Table 1 polymers-13-02724-t001:** Mass balance for 5000 PMMA scraps tons/year plant, in case of Cast and Mixed PMMA scraps.

Feedstock/Product/Waste	Cast PMMA Scraps (90 wt % yield)	Mixed PMMA Scraps (78 wt % yield)
PMMA scraps (tons/y)	5000	5000
Natural Gas (tons/y)	106	80
Heavy Ends (tons/y)	119	411
Solid Waste (tons/y)	100	200
r-MMA (tons/y)	4512	3889

**Table 2 polymers-13-02724-t002:** Historical data of first of a kind (FOAK) or nth of a kind (NOAK) plastic to liquid plants.

Company	Capacity (ktons/year)	Investment (M US$)	Year	Project Type	Process Type
Cyclix [38]	33	80	2020	NOAK	Plastic to Oil
Plastic Energy [36,39]	5	20	2014/15	FOAK	Plastic to Oil
Plastic Energy [36]	20–25	35	2019	NOAK	Plastic to Oil
Renewlogy [40]	3.3	4	2018	NOAK	Plastic to Fuel
Renew Phoenix [41,42]	3.3	5.5	2019	NOAK	Plastic to Fuel
QM Recycled Energy [43]	10	9	2018	FOAK	Plastic to Oil
QM Recycled Energy/Biofabric [44]	1.6	2	2020	FOAK	Plastic to Oil
Quantafuel [45]	20	17.5	2018	FOAK	Plastic to Fuel
Quantafuel [46]	25	18	2020	NOAK	Plastic to Fuel
Quantafuel/Vitol [45,47]	100	75	2020	NOAK	Plastic to Fuel
Recycling Technologies [48]	7	7.7	2018	FOAK	Plastic to Oil
Carbo Hydro Transformation (seems to have now disappeared)	8	2	2018	FOAK	Plastic to Fuel

**Table 3 polymers-13-02724-t003:** Scenarios and prices overview.

Scenario	1	2	3	4	5	6	7	8	9
**Scraps**	Mixed	Mixed	Mixed	Mixed	Cast	Cast	Mixed 0 price	Mixed	Mixed 0 price
**r-MMA (price)**	Virgin	Virgin	20% premium	20% discount	Virgin	20% premium	20% discount	20% discount	20% discount
**Capital Investment**	10 M $	7 M $ (30% subsidies)	10 M $	1.3 M $ 20 years lifetime	7 M $	7 M $	10 M $	1.3 M 10 years lifetime	7 M $ (30% subsidies)
**Feedstock/Product/Waste**		**Distribution Law**		**Mean, Deviation**		**[5%, 95%] Probability ($/ton)**			
PMMA Mixed		Log-Normal		[5.00, 0.24]		[100, 220]			
PMMA Cast		Log-Normal		[6.00, 0.20]		[305, 590]			
Natural Gas		Log-Normal		[5.90, 0.40]		[190, 705]			
Heavy Ends		Gaussian		[150, 50]		[70, 240]			
Solid Waste		Gaussian		[250, 50]		[165, 300]			
r-MMA 20% discount		Log-Normal		[7.40, 0.14]		[1290, 2080]			
r-MMA Virgin		Log-Normal		[7.65, 0.14]		[1655, 2670]			
r-MMA 20% premium		Log-Normal		[7.84, 0.14]		[2000, 3225]			

**Table 4 polymers-13-02724-t004:** Proposed correlation matrix for the future, based on historical data and expert judgement. Green stands for highly correlated, while red for lowly correlated.

	PMMA Scraps	Natural Gas	Heavy Ends	Solid Waste	r-MMA
PMMA Scraps	1				
Natural Gas	0.2	1			
Heavy ends	0.05	0.5	1		
Solid Waste	0.05	0.05	0.8	1	
r-MMA	0.8	0.4	0.4	0.05	1

**Table 5 polymers-13-02724-t005:** Ranking of Scenarios according to different criteria/expected outcome.

Criteria	Ranking (Scenarios)	Conclusions
Pay-back time (from short to long)	**Scenarios 8/4** Scenario 6 Scenarios 2/3/5 Scenario 9 Scenario 1	Cheap technology/second-hand equipment plant pay back in the shortest time.
NPV after 10 y of operation (highest first)	**Scenarios 3** Scenario 2/5 Scenarios 8 Scenario 9 Scenario 1	After 10 y of operation, high-quality r-MMA, coupled with expensive technology pays off more than cheap plants, or low-quality scraps at zero price. If the market can accept both high and low purity product, companies should target high-quality r-MMA.
Probability of losing money after 10 y of operation (from low to high)	**Scenario 6** Scenario 2/3/5 Scenario 9 Scenario 1 Scenario 7	Despite feedstock available for free, Scenario 7 has more probability of losses than the base case Scenario 1. The better the quality of the scraps/product, the lower the probability to lose money becomes.

## Data Availability

The authors confirm that the data supporting the findings of this study are available within the article.

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
