# Peer review of "Risk Analysis on PMMA Recycling Economics"

_polymers, 2021, doi:10.3390/polym13162724_

Round 1

Reviewer 1 Report

Dear Editor,

I have read the manuscript entitled: “Risk Analysis on PMMA Recycling Economics” and the authors have done a truly significant work on the Risk Analysis on PMMA Recycling Economics, but there are some format mistakes and inconsistence throughout the manuscript:

Pg. 1, line 7: PolyMethyl MethAcrylate (PMMA) -  should be - Poly(methyl methacrylate) (PMMA), as in line 32

Pg. 1, line 15: MMA quality have the main -  should be - MMA quality has the main

Pg. 1, line 27: quality product -  should be - quality products

Pg. 2, line 58: represent a relatively small volumes -  should be - represent a relatively small volume

Pg. 2, line 83: properly there are no contamination -  should be - properly there is no contamination - 

Pg. 4, line 180: aim to start with -  should be - : aim start with

Pg. 8, lines 250,251: on the third -  should be – in the third;  on the fourth -  should be – in the fourth; 

Pg. 8, lines 270: while in 2019 was 607.5 -  should be – while in 2019 it was 607.5

Pg. 8, lines 291: 3-4” for chemist, -  should be – 3-4” for chemists

Pg. 10, lines 327: tion are a good starting -  should be – tion is a good starting

Pg. 11, lines 359: 8000 traded in within Europe -  should be – 8000 traded within Europe

Pg. 18, lines 524: When that will happen, -  should be – When that will happens,

Pg. 18, lines 532: and the purchasing power increases. -  should be – and their purchasing power increases.

Author Response

We thank th reviewer for his comments and careful reading.

Pg. 1, line 7: PolyMethyl MethAcrylate (PMMA) -  should be - Poly(methyl methacrylate) (PMMA), as in line 32.

  • We thank the reviewer for the comment, and we revised the text as suggested. The manuscript now reads: Poly(methyl methacrylate)

Pg. 1, line 15: MMA quality have the main -  should be - MMA quality has the main

  • We thank the reviewer for the comment, but the sentence has capital investment and MMA quality as subject, so have seems the appropriate verb.

Pg. 1, line 27: quality product -  should be - quality products

  • We thank the reviewer for the comment, and we revised the text as suggested. The manuscript now reads: products

Pg. 2, line 58: represent a relatively small volumes -  should be - represent a relatively small volume

  • We thank the reviewer for the comment, and we revised the text as suggested. The manuscript now reads: volume

Pg. 2, line 83: properly there are no contamination -  should be - properly there is no contamination – 

  • We thank the reviewer for the comment, and we revised the text as suggested. The manuscript now reads: there is no contamination

Pg. 4, line 180: aim to start with -  should be - : aim start with

  • We thank the reviewer for the comment, and we revised the text as suggested. The manuscript now reads: aim to start with

Pg. 8, lines 250,251: on the third -  should be – in the third;  on the fourth -  should be – in the fourth; 

  • We thank the reviewer for the comment, and we revised the text as suggested. The manuscript now reads: in the fourth … in the fifth…

Pg. 8, lines 270: while in 2019 was 607.5 -  should be – while in 2019 it was 607.5

  • We thank the reviewer for the comment, and we revised the text as suggested. The manuscript now reads: in 2019 it was …

Pg. 8, lines 291: 3-4” for chemist, -  should be – 3-4” for chemists

  • We thank the reviewer for the comment, and we revised the text as suggested. The manuscript now reads: for chemists

Pg. 10, lines 327: tion are a good starting -  should be – tion is a good starting

  • We thank the reviewer for the comment, and we revised the text as suggested. The manuscript now reads: is a good starting

Pg. 11, lines 359: 8000 traded in within Europe -  should be – 8000 traded within Europe

  • We thank the reviewer for the comment, and we revised the text as suggested. The manuscript now reads: traded within Europe

Pg. 18, lines 524: When that will happen, -  should be – When that will happens,

  • We thank the reviewer for the comment, but we believe that “when that will happen” is the correct grammatical form

Pg. 18, lines 532: and the purchasing power increases. -  should be – and their purchasing power increases.

  • We thank the reviewer for the comment, and we revised the text as suggested. The manuscript now reads: and their purchasing power increases

Reviewer 2 Report

The manuscript entitled: "Risk Analysis on PMMA recycling economics" is an interesting work summarizing many of the important economic factors that govern the recycling of a common plastic in everyday life, especially after the appearance of COVID19. It is a work that is worth getting published; however, a few points need a critical check

  • please write in abstract Poly(Methyl MethAcrylate).
  • What is the highest purity grade of the MMA that is generated by the process? Can the authors provide the maximum clean grade of the monomer?
  • Do the authors consider a model method for the utilization of the monomer for a new polymerization?
  • How much more energy efficient can the process be done by utilising energy like solar or wind energy?
  • To what extent can the process be further optimized by the utilization of membrane technology? An example of a recovery process for organic solvents is presented here: Processes 20219(5), 729; https://doi.org/10.3390/pr9050729. It would be nice to do a correlation with such a closed-loop process.
  • Last but not least, many types of PMMA plastic do contain additives such as plasticizers, colours, etc. Is it possible to consider recycling these products by e.g.m fractional destination or membrane destination? Also, what is the impact of their small content on the recovered MMA?

Author Response

We thank the reviewer for his comments. Each of them are addressed below.

1) please write in abstract Poly(Methyl MethAcrylate).

We thank the reviewer, and we addressed this point in the responses to Reviewer 1.

2) What is the highest purity grade of the MMA that is generated by the process? Can the authors provide the maximum clean grade of the monomer?

  • We analyzed the risks of a PMMA recycling plant in the European framework. In our view of the future, the process should aim at the highest purity to be competitive in the EU market. This translates, for instance, in a r-MMA purity above 99 wt% after purification, which is what the MMAtwo consortium has achieved so far. With the technology developed in the MMAtwo Project 99.8 wt% pure, or virgin like purityhas been achieved. However, that’s at the expense of the yield. A high purity regenerated MMA, with a premium as discussed in our paper, should be obtained at a purity not achieved with the competitors, which means more than 99 % and probably more than 99.5 %.

3) Do the authors consider a model method for the utilization of the monomer for a new polymerization?

  • We thank the reviewer for the comment, however we did not consider such a model method, since it was outside the scope of our analysis. What we see from the market, is that even purities as low as 93 % are being repolymerized in India and in Brazil. However, in the European context, it is most likely that purities above 98.5 % are needed. At that level both cast and extrusion/injection polymers are produced. Other discussions on conditions of repolymerization are beyond the scope of the publication.

4) How much more energy efficient can the process be done by utilising energy like solar or wind energy?

  • In the current process configuration, we assumed the use of Natural gas to provide heat. It could be bio-sourced as biogas. At the same time, the energy integration system (Figure 2), recovers steam for the rotative equipment of the plant (pumps, stirred reactor, etc.). We could, of course, switch completely to electricity, and power the plant with energy from renewable sources. Solar and wind electricity, are strongly depending on climate conditions, so they cannot be the sole source of energy in the process which is analyzed as a continuous process. If we would analyze it as a batch process, operating only when the sustainable energy is available, we would have to invest much more and operate the plant only from time to time which would not improve the economics of the plant.
  • Rather than the energy efficiency, this would impact the environmental footprint of the plant. Solar or wind electricity would cut the CO2 footprint, but the regenerated MMA has already a much better carbon footprint than the virgin MMA – in the MMAtwo project we have shown that a 70 % reduction is achievable. So the current focus is to identify how to make it in an economic way

5) To what extent can the process be further optimized by the utilization of membrane technology? An example of a recovery process for organic solvents is presented here: Processes 20219(5), 729; https://doi.org/10.3390/pr9050729. It would be nice to do a correlation with such a closed-loop process.

  • We thank the reviewer for the comment. The purification of monomers like MMA is challenging as it tends to repolymerize easily. In the conventional virgin MMA purification, the membrane-based processes are not commonly used and that’s the same for other acrylates, in part because of that tendency to polymerize in absence of stabilizers. The separation of water by a pervaporation membrane would be justified if there is a significant amount of water in the products, but in our model we assumed that there would not be a washing step before the distillation, so the water content would be limited to the solubility of water in MMA, which is around 1 wt %. So much simpler solutions than a membrane separation would be implemented for water removal. For the other impurities in crude MMA, the purification technologies would have to be assessed on a case by case. Finally the purification of MMA by distillation is not so energy intensive, since the regenerated MMA has an equivalent energy consumption about 70 % lower than virgin MMA. Energy saving would not be the prime target to promote PMMA recycling, but as demonstrated it should be the improvement of product quality and the reduction of the Capital cost.
  •  

6) Last but not least, many types of PMMA plastic do contain additives such as plasticizers, colours, etc. Is it possible to consider recycling these products by e.g.m fractional distillation or membrane distillation? Also, what is the impact of their small content on the recovered MMA?

  • We thank the reviewer for the comment. In the current analysis, we assumed all the additives to end up as a solid waste in the depolymerization step. The nature and content of the plasticizer depends on the final application of the polymer. Therefore, unless the process only targeted post-industrial waste, from a given tracked source (e.g. directly from a PMMA producer), it would be difficult to know exactly which additives are present. In the MMAtwo experience, and in the existing process we benchmarked, these additives mostly end up as solid waste in the depolymerization step, therefore affecting the global r-MMA yield. Some comonomers, for instance Ethyl Acrylate (EA), are present in a particular grade of PMMA (i.d. extrusion or injection PMMA). Ethyl Acrylate has the same boiling point than MMA, and cannot be separated by distillation. That’s indeed a special case where a membrane separation could make sense, but the chemical affinity of the 2 molecules is extremely close. A regenerated MMA, which contains Ethyl Acrylate and/or Methyl Acrylate can still be repolymerized, so those impurities are not detrimental to the potential application. In order to keep the cost as low as possible, only the necessary purifications should be taken into account.
  • A short discussion has been added in the text.

Reviewer 3 Report

The authors analyze the challenges of PMMA recycling through depolymerization generating MMA. The manuscript presents an overview of the possibility of PMMA recycling. It is a good document from a technological point of view.

Point 1. There is no talk of the by-products that can be generated by establishing a recycling company in the scenarios proposed, specifically in annual estimates. Is it profitable for the environment?

The solid wastes mentioned in Table 3, what type are they?

Point 2. The document has many grammatical errors.

Author Response

We thank the reviewer for his comments.

 1). There is no talk of the by-products that can be generated by establishing a recycling company in the scenarios proposed, specifically in annual estimates. Is it profitable for the environment?

  • We thank the reviewer for the comment. In fact, the by-products generated are the “solid waste” and “heavy ends” of Table 1. Depending on the type of feedstock, we reported the annual generated amount. In our analysis, we assumed a waste treatment cost of 150-250 USD/ton (Figure 11) for these by-products. After analyzing the European disposal (landfill + tipping fee) for different countries in the continent (Section 2.3.3), we assumed that in the near future the disposal cost will level up to the highest in the continent. If the PMMA scraps come from a post-industrial source, the by-products might be still clean enough to find some market as fuel (Heavy ends), or in the heavy industry (solid waste). However, if the scraps are from post-consumer materials, for instance from 20 years ago, the by-products cannot enter the market and may contain hazardous substances and substances of very high concerns. Therefore, as opposed to plant built in India, Brazil, Indonesia, our plant properly disposes of such polluted streams according to the most stringent European standards. To highlight this aspect, we added the sentence :

In this way, we assume to treat the plant by-products according to the highest European standards in terms of waste disposal, as opposed of what happens in existing lower quality PMMA recycling plant in developing countries (e.g. India, Brazil, etc.).

2) The solid wastes mentioned in Table 3, what type are they?

  • The solid wastes are charcoal like, or solid fillers/additives present in the original polymer in small quantity, based on the polymer final application. They may contain some residues of pigments, crosslinking agents, fillers, other polymers residues;..

3). The document has many grammatical errors.

  • We thank the reviewer for the comment. The document underwent a thorough revision, and we corrected all the errors we could find. The corrections appear highlighted in the track mode.